# Prioritizing Samples in Reinforcement Learning with Reducible Loss

**Shivakanth Sujit**
Mila, Quebec AI Institute, ÉTS Montréal
shivakanth.sujit@gmail.com

**Somjit Nath**
Mila, Quebec AI Institute, ÉTS Montréal
somjitnath@gmail.com

**Pedro H.M. Braga**
Mila, Quebec AI Institute, ÉTS Montréal, Universidade Federal de Pernambuco
pedromagalhaes.hb@gmail.com

**Samira Ebrahimi Kahou**
Mila, Quebec AI Institute, ÉTS Montréal, CIFAR AI Chair
samira.ebrahimi.kahou@gmail.com

## Abstract

Most reinforcement learning algorithms take advantage of an experience replay buffer to repeatedly train on samples the agent has observed in the past. Not all samples carry the same amount of significance and simply assigning equal importance to each of the samples is a naive strategy. In this paper, we propose a method to prioritize samples based on how much we can learn from a sample. We define the *learn-ability* of a sample as the steady decrease of the training loss associated with this sample over time. We develop an algorithm to prioritize samples with high learn-ability, while assigning lower priority to those that are hard-to-learn, typically caused by noise or stochasticity. We empirically show that across multiple domains our method is more robust than random sampling and also better than just prioritizing with respect to the training loss, i.e. the temporal difference loss, which is used in prioritized experience replay. The code to reproduce our experiments can be found here.

## 1   Introduction

Deep reinforcement learning has shown great promise in recent years, particularly with its ability to solve difficult games such as Go [Silver et al., 2016], chess [Silver et al., 2018], and Atari [Mnih et al., 2015]. However, online Reinforcement Learning (RL) suffers from sample inefficiency because updates to network parameters take place at every time-step with the data being discarded immediately. One of the landmarks in the space of online RL learning has been Deep Q Networks (DQN) [Mnih et al., 2015], where the agent learns to achieve human-level performance in Atari 2600 games. A key feature of that algorithm was the use of batched data for online learning. Observed transitions are stored in a buffer called the *experience replay* [Lin, 2004], from which one randomly samples batches of transitions for updating the RL agent.

Instead of randomly sampling from the experience replay, we propose to sample based on the *learn-ability* of the samples. We consider a sample to be learnable if there is a potential for reducing the agent's loss with respect to that sample. We term the amount by which we can reduce the loss of a sample to be its *reducible loss* (ReLo). This is different from vanilla prioritization in Schaul et al.

[2016] which just assigns high priority to samples with high loss, which can potentially lead to repeated sampling of data points which can not be learned from due to noise.

In our paper, we first briefly describe the current methods for prioritization while sampling from the buffer, followed by an intuition for reducible loss in reinforcement learning. We demonstrate the performance of our approach empirically on the DeepMind Control Suite [Tassa et al., 2018], OpenAI Gym, MinAtar [Young and Tian, 2019] and Arcade Learning Environment [Bellemare et al., 2013] benchmarks. These experiments show how prioritizing based on the reducible loss is a more robust approach compared to just the loss term [Schaul et al., 2016] used in Hessel et al. [2017] and that it can be integrated without adding any additional computational complexity.

## 2 Background and Related Work

In Reinforcement Learning (RL), an agent is tasked with maximizing the expected total reward it receives from an environment via interaction with it. This problem is formulated using a Markov Decision Process (MDP) [Bellman, 1957] that is described by $< \mathcal{S}, \mathcal{A}, \mathcal{R}, \mathcal{P} >$, where $\mathcal{S}, \mathcal{A}, \mathcal{R}$ and $\mathcal{P}$ represent the state space, the action space, the reward function, and the transition function of the environment, respectively. The objective of RL is to learn an optimal policy $\pi^*$, which is a mapping from states to actions that maximizes the expected discounted sum of rewards it receives from the environment, that is

$$\pi^* = \underset{\pi}{\operatorname{argmax}} \, \mathbb{E}_\pi \left[ \sum_{t=0}^{\infty} \gamma^t r_t | S_t = s, A_t = a \right], \tag{1}$$

where $\gamma \in [0, 1]$ is the discount factor. Action value methods obtain a policy by learning the action value ($Q^\pi(s_t, a_t)$) of a policy which is the expected return by taking action $a_t$ in state $s_t$ and then following the policy $\pi$ to choose further actions. This is done using the Bellman equation, which defines a recursive relationship in terms of the $Q$ value function, as follows

$$Q^\pi(s_t, a_t) = r_t + \gamma \max_a Q^\pi(s_{t+1}, a) \tag{2}$$

The difference between the left and right sides of Eq. 2 is called the temporal difference error (TD error), and $Q$ value methods minimize the TD error of the learned $Q$ function $Q^\theta$ (implemented as a neural network) using stochastic gradient descent. That is, the loss for the $Q$ network is

$$L_\theta = (Q^\theta(s_t, a_t) - (r_t + \gamma \max_a Q^\theta(s_{t+1}, a)))^2. \tag{3}$$

We can then use the $Q$ value to implicitly represent a policy by choosing actions with high $Q$ values. While this is easy in discrete control tasks which have a small action space, it can be difficult in continuous action spaces because finding the action that maximizes the $Q$ value can be an optimization problem in itself. This can be computationally expensive to do at every instant, so recent methods alleviate this problem through an actor network $\mu_\theta$ that learns the action that produces the maximum $Q$ value through stochastic gradient ascent, that is

$$\mu_\theta = \underset{\theta}{\operatorname{argmax}} \, Q^\theta(s_t, \mu_\theta(s_t)). \tag{4}$$

The loss for the $Q$ network in Eq. 3 is then modified so that the argmax is evaluated using the actor network,

$$L_\theta = (Q^\theta(s_t, a_t) - (r_t + \gamma \, Q^\theta(s_{t+1}, \mu_\theta(s_t))))^2 \tag{5}$$

### 2.1 Experience Replay

Online RL algorithms perform updates immediately after observing a transition. However, these not only make learning inefficient but can cause issues in training due to high correlation between recent transitions. To eliminate this problem, Lin [2004] introduced experience replay, which stores the observed transitions and provides an interface to sample batches of transitions. This has been successfully used in DQN [Mnih et al., 2015] to play Atari 2600 games. Since Eqs. 3 and 5 do not require that the states and actions are generated from the current policy, algorithms trained this way are called off-policy RL algorithms. During training, data is collected from the environment and stored in a replay buffer from which mini-batches are sampled to be trained on.

A naive method of sampling is to uniformly sample all data in the buffer, however, this is inefficient because not all data is necessarily equally important. Schaul et al. [2016] proposes Prioritized Experience Replay (PER), that samples points with probabilities proportional to their TD error – which has been shown to have a positive effect on performance by efficiently replaying samples that the model has not yet learned, i.e., data points with high TD error. Each transition in the replay buffer is assigned a priority $p_i$, and the transitions are sampled based on this priority. To ensure that data points, even with low TD error, are sampled sometimes by the agent, instead of greedy sampling based on TD error, the replay buffer in PER stochastically samples points with probability $P_i$.

$$P_i = \frac{p_i^{\alpha}}{\sum_j p_j^{\alpha}} \tag{6}$$

where $\alpha \in [0, 1)$ is a hyper-parameter introduced to smoothen out very high TD errors. Setting $\alpha$ to 0 makes it equivalent to uniform sampling. Since sampling points non-uniformly changes the expected gradient of a mini-batch, PER corrects for this by using importance sampling (IS) weights $w$

$$w_i = \left( \frac{p_{\text{uniform}}}{P_i} \right)^{\beta} \tag{7}$$

where $\beta \in [0, 1]$ controls the amount by which the change in gradient should be corrected and $p_{\text{uniform}} = \frac{1}{N}$ where $N$ is the number of samples in the replay buffer. The loss attributed to each sample is weighed by the corresponding $w_i$ before the gradient is computed. In practice, $\beta$ is either set to $0.5$ or linearly annealed from $0.4$ to $1$ during training.

## 2.2 Target Networks

In Eqs. 3 and 5, the target action value depends not only on the rewards but also on the value of the next state, which is not known. So, the value of the next state is approximated by feeding the next state to the same network used for generating the current $Q$ values. As mentioned in DQN [Mnih et al., 2015], this leads to a very unstable target for learning due to the frequent updates of the $Q$ network. To alleviate this issue, Mnih et al. [2015] introduce target networks, where the target $Q$ value is obtained from a lagging copy of the $Q$ network used to generate the current $Q$ value. This prevents the target from changing rapidly and makes learning much more stable. So Eq. 3 can be suitably modified to

$$L_\theta = (Q^\theta(s_t, a_t) - (r_t + \gamma \max_a Q^{\bar{\theta}}(s_{t+1}, a)))^2 \tag{8}$$

respectively, where $\bar{\theta}$ are the parameters of the target network, which are updated at a low frequency. Mnih et al. [2015] copies the entire training network $\theta$ to the target network, whereas Haarnoja et al. [2018] performs a soft update, where the new target network parameters are an exponential moving average (with a parameter $\tau$) of the old target network parameters and the online network parameters.

## 2.3 Reducible Loss

The work of Mindermann et al. [2022] proposes prioritized training for supervised learning tasks based on focusing on data points that reduce the model's generalization loss the most. Prioritized training keeps a held-out subset of the training data to train a small capacity model, $\theta_{ho}$ at the beginning of training. During training, this hold-out model is used to provide a measure of whether a data point could be learned without training on it. Given training data $(x_i, y_i) \in \mathcal{D}$, the loss of the hold-out model's prediction, $\hat{y}_{i_{ho}}$ on a data point $x_i$ could be considered an estimate of the remaining loss after training on datapoints other than $(x_i, y_i)$, termed the *irreducible loss*. This estimate becomes more accurate as one increases the size of the held-out dataset. The difference between the losses of the main model, $\theta$, and the hold-out model on the actual training data is called the *reducible loss*, $L_r$ which is used for prioritizing training data in mini-batch sampling. $L_r$ can be thought of as a measure of information gain by also training on data point $(x, y)$.

$$L_r = \text{Loss}(\hat{y} \mid x, \theta) - \text{Loss}(\hat{y} \mid x, \theta_{ho}) \tag{9}$$

## 2.4 Prioritization Schemes

Alternate prioritization strategies have been proposed for improvements in sample efficiency. Sinha et al. [2020] proposes an approach that re-weights experiences based on their likelihood under the stationary distribution of the current policy in order to ensure small approximation errors on the value function of recurring seen states. Lahire et al. [2021] introduces the Large Batch Experience Replay (LaBER) to overcome the issue of the outdated priorities of PER and its hyperparameter sensitivity by employing an importance sampling view for estimating gradients. LaBER first samples a large batch from the replay buffer then computes the gradient norms and finally down-samples it to a mini-batch of smaller size according to a priority. Kumar et al. [2020] presents Distribution Correction (DisCor), a form of corrective feedback to make learning dynamics more steady. DisCor computes the optimal distribution and performs a weighted Bellman update to re-weight data distribution in the replay buffer. Inspired by DisCor, Regret Minimization Experience Replay (ReMERN) [Liu et al., 2021] estimates the suboptimality of the Q value with an error network. Yet, Hong et al. [2022] uses Topological Experience Replay (TER) to organize the experience of agents into a graph that tracks the dependency between Q-value of states.

## 3    Reducible Loss for Reinforcement Learning

While PER helps the agent to prioritize points that the model has not yet learned based on high TD error, we argue that there are some drawbacks. Data points could have *high* TD error because they are noisy or not learnable by the model. It might not be the case that a data point with high TD error is also a sample that the model can actually learn or get a useful signal from. In supervised learning, a known failure condition of loss based prioritization schemes is when there are noisy points which can have high loss but are not useful for repeated training [Hu et al., 2021]. Instead of prioritization based on the TD error, we propose that the agent should focus on samples that have higher *reducible* TD error. This means that instead of the TD error, we should use a measure of how much the TD error can be potentially decreased, as the priority $p_i$ term in Eq. 6. We contend that this is better because it means that the algorithm can avoid repeatedly sampling points that the agent has been unable to learn from and can focus on minimizing error on points that are learnable, thereby improving sample efficiency. Motivated by prioritized training in supervised learning, we propose a scheme of prioritization tailored to the RL problem.

In the context of supervised learning, learn-ability and reducible loss for a sample are well-defined as one has access to the true target. However, in RL, the true target is approximated by a bootstrapped target (TD methods) or the return obtained from that state (Monte Carlo). For policy evaluation with a fixed policy, the true value function can be obtained, however, since we are interested in control, it will be computationally intensive to capture the true value function with every change in policy as in policy iteration. Thus, to determine the learn-ability of a sample, we need access to how the targets of the sample behave and how it changes across time. Since the concepts of a hold-out dataset or model are ill-defined in the paradigm of RL, we replace them with a moving estimate of the targets. Unlike in supervised learning, where we draw i.i.d. batches from a fixed training set, the training data in RL are not i.i.d since they are generated by a changing policy. So the holdout model would need to be updated from time to time. Thus, in $Q$ learning-based RL methods, a good proxy for the hold-out model is the target network used in the Bellman update in Eq. 8. Since the target network is only periodically updated with the online model parameters, it retains the performance of the agent on older data which are trained with outdated policies. Schaul et al. [2022] demonstrates how the policies keep changing with more training even when the agent receives close to optimal rewards. Thus, the target network can be easily used as an approximation of the hold out model that was not trained on the new samples. Therefore, we define the Reducible Loss (ReLo) for RL as the difference between the loss of the data point with respect to the online network (with parameters $\theta$) and with respect to the target network (with parameters $\bar{\theta}$). So the Reducible Loss (ReLo) can be computed as

$$\text{ReLo} = L_\theta - L_{\bar{\theta}} \tag{10}$$

There are similarities between ReLo as prioritization scheme in the sampling behavior of low priority points when compared to PER. Data points that were not important under PER, i.e. they have low $L_\theta$, will also remain unimportant in ReLo. This is because if $L_\theta$ is low, then as per Eq. 10, ReLo will also be low. This ensures that we retain the desirable behavior of PER, which is to not repeatedly sample points that have already been learned.

However, there is a difference in sampling points that have high TD error. PER would assign high priority to data points with high TD error, regardless of whether or not those data points are noisy or unlearnable. For example, a data point can have a high TD error which continues to remain high even after being sampled several times due to the inherent noise of the transition itself, but it would continue to have high priority with PER. Thus, PER would continue to sample it, leading to inefficient learning. But, its priority should be reduced since there might be other data points that are worth sampling more because they have useful information which would enable faster learning. The ReLo of such a point would be low because both $L_\theta$ and $L_{\bar{\theta}}$ would be high. In case a data point is forgotten, then the $L_\theta$ would be higher than $L_{\bar{\theta}}$, and the ReLo would ensure that these points are revisited. Thus ReLo can also help to overcome forgetting.

## 3.1    Implementation

The probability of sampling a data point is related to the priority through Eq. 6 and requires the priority to be non-negative. Since $Q$ value methods use the mean-squared error (MSE) loss, the priority is guaranteed to be non-negative. However, ReLo computes the difference between the MSE losses and it does not have the same property. This value can go to zero when the target network is updated with the main network using hard updates. However, it quickly becomes non-zero after one update. So, we should create a mapping $f_{map}$ for the ReLo error that is monotonically increasing and non-negative for all values. In practice, we found that clipping the negative values to zero, followed by adding a small $\epsilon$ to ensure samples had some minimum probability, worked well. That is, $p_i = max(\text{ReLo}, 0) + \epsilon$. This is not the only way we can map the negative values and we have studied one other mapping in the supplementary material. Note also that during initial training when the agent sees most of the points for the first time, it would assign low priority to all depending on the value of ReLo and the sampling would be close to random. However, the priority will never go to zero because of $\epsilon$, so we will always have a non-zero probability of drawing the sample. Once it is sampled and the loss goes down, then the priority would again increase as per ReLo and it would prioritize such points until the ReLo becomes low again. Thus, it kind of achieves the best of both worlds, where initially with less information we sample uniformly, but once ReLo increases we sample points more frequently.

ReLo is not computationally expensive since it does not require any additional training. It only involves one additional forward pass of the states through the target network. This is because the Bellman backup (i.e., the right hand side of Eq. 2) is the same for $L_\theta$ and $L_{\bar{\theta}}$. The only additional term that needs to be computed for ReLo is $Q_{tgt}(s_t, a_t)$ to compute $L_{\bar{\theta}}$.

---

**Algorithm 1** Computing ReLo for prioritization

---

Given off-policy algorithm $A$ with loss function $L^{alg}$, online $Q$ network parameters $\theta$, target $Q$ network parameters $\bar{\theta}$, replay buffer $B$, max priority $p_{max}$, ReLo mapping $f_{map}$, epsilon priority $\epsilon$, training timesteps $T$, gradient steps per timestep $T_{grad}$, batch size $b$.

**for** t in 1, 2, 3, . . . $T$ **do**

    Get current state $s_t$ from the environment

    Compute action $a_t$ from the agent

    Store the transition $< s_t, a_t, r_t, s_{t+1} >$ in the replay buffer $B$ with priority $p_{max}$.

    **for** steps in 1, 2, 3, . . . $T_{grad}$ **do**

        Sample minibatch of size $b$ from replay buffer

        Compute the loss $L_\theta^{alg}$ and update the agent parameters $\theta$

        Compute $L_{\theta tgt}^{alg}$ and calculate ReLo as per Eq. 10

        Update priorities of samples in mini-batch with the new ReLo values as $f_{map}(\text{ReLo}_i) + \epsilon$

    **end for**

    Update target network following the original RL algorithm $A$

**end for**

---

In our implementation, we saw a negligible change in the computational time between PER and ReLo. ReLo also does not introduce any additional hyper-parameters that need to be tuned and works well with the default hyper-parameters of $\alpha$ and $\beta$ in PER. An important point to note is that ReLo does not necessarily depend on the exact loss formulation given in Eq. 8 and can be used with the loss function $L_\theta^{alg}$ of any off-policy $Q$ value learning algorithm. In order to use ReLo, we only have to additionally

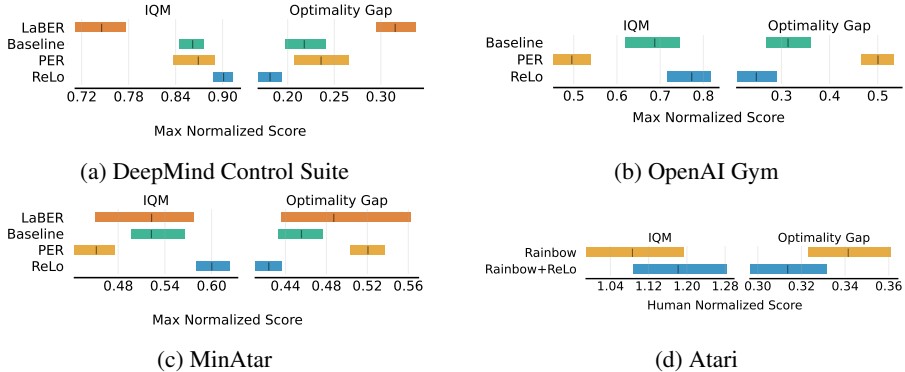

Figure 1: Metrics aggregated across each benchmark based on proposed metrics from Agarwal et al. [2021]. 10 seeds per environment-algorithm pair.

compute $L^{alg}$ with respect to the target network parameters $\bar{\theta}$. If the loss is just the mean square error, then ReLo can be simplified and can be represented by the difference between $Q_\theta$ and $Q_{\bar{\theta}}$. But other extensions to off policy Q learning methods modify this objective, for example distributional learning [Bellemare et al., 2017] minimizes the KL divergence and the difference between two KL divergences can not be simplified the same way. To make ReLo a general technique that can be used across these methods, we define it in terms of $L_\theta$ and $L_{\bar{\theta}}$. Our experiments also show that ReLo is robust to the target network update mechanism, whether it is a hard copy of online parameters at a fixed frequency (as in DQN [Mnih et al., 2015], and Rainbow [Hessel et al., 2017]) or if the target network is an exponential moving average of the online parameters (as in Soft Actor Critic [Haarnoja et al., 2018]).

## 4 Experimental Results

### 4.1 GridWorld Experiments

The major goal of these Gridworld studies is to draw attention to two PER issues: subpar performance when faced with stochasticity and forgetting when faced with many tasks. We also highlight how ReLo can potentially solve both issues.

### 4.1.1 Pitfalls of TD Error Prioritization

To illustrate the potential downsides of using PER with TD error prioritization, we created a $7 \times 7$ empty gridworld with a single goal. The agent always spawns at the top left corner and it has to reach the goal at the bottom right for a reward of $+2$. The reward is zero everywhere else except, near the center of the grid, where there is another state with a reward sampled uniformly from $[-0.5, 0.5]$[1] The presence of this *critical* point would mean that during the initial phase of training, PER would keep prioritizing transitions to that state due to the unpredictable reward and the corresponding high TD error, and fail to learn about the true goal. Given a sufficient number of samples, all algorithms would ultimately converge to the optimal policy. The ReLo criterion, however, does not prioritize unlearnable points and prioritizes other points where it can receive a larger reward. Fig. 2 shows that PER samples this unlearnable point very often and with limited experience is unable to solve the task. In contrast, both uniform experience replay and ReLo do not over-sample this transition and thus can perform better in this scenario. This is further affirmed by the number of updates each algorithm makes in the neighborhood of the *unlearnable* point as visualized in Fig. 2. Prioritization using ReLo enables the agent to sample more from the transitions leading to the main reward which explains its better performance.

---

[1]A detailed description of the environment is given in the supplementary material.

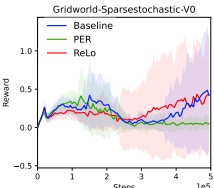 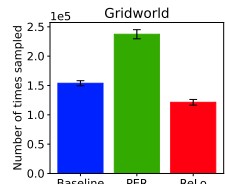

Figure 2: Left: The rewards obtained after 50K episodes across 50 runs in the gridworld domain with an unlearnable point. Right: Number of times a transition containing the *unlearnable* state and its neighbors were sampled.

### 4.1.2 Mitigating forgetting with ReLo

To study how ReLo can help reduce forgetting in RL, we design a multi task version of the gridworld. We create a $6 \times 6$ gridworld consisting of two rooms A and B and a goal state in each room. Task A is defined as starting in Room A and reaching the goal state in Room A, and Task B is defined similarly for Room B. There is a single gap in between the rooms, allowing the agent to explore both rooms but a time limit is introduced so that the agent can not reach both goals in one episode. During training, for the first 100k steps, the agent starts in Room A with access to Room B blocked. So during this stage, the agent observes only Task A. After 100K steps, the agent starts only in Room B, thereby no longer collecting data about Task A and must retain its ability on the task by replaying relevant transitions from the buffer. We train three agents, a baseline DQN agent, a PER agent and a ReLo agent and monitor performance on both tasks during training. We provide average success rates over 60 seeds after 1M steps in Table 1 and Appendix J. It shows how ReLo exhibits the least degradation in performance on Task A compared to the other baselines. From the training curves, we can see that there is a clear drop off in performance on Task A after 100K steps when the agent can no longer actively collect data on the task. However the ReLo agent exhibits least degradation in Task A while also outperforming the baseline and PER agent on Task B. This experiment clearly shows how ReLo helps the agent replaying relevant data points that could have been forgotten.

Table 1: Performance on Forgetting Task. Confidence intervals in brackets.

| Algorithm | Task A | Task B |
|---|---|---|
| Uniform | 0.43 (0.32, 0.54) | 0.40 (0.29, 0.51) |
| PER | 0.29 (0.19, 0.39) | 0.26 (0.16, 0.36) |
| ReLo | **0.63 (0.52, 0.74)** | **0.74 (0.65, 0.84)** |

### 4.2 Comparison of PER and ReLo

We study the effectiveness of ReLo on several continuous and discrete control tasks. For continuous control, we evaluate on 9 environments from the DeepMind Control (DMC) benchmark [Tassa et al., 2018] as they present a variety of challenging robotic control tasks with high dimensional state and action spaces. We also include 3 environments from the OpenAI Gym benchmark for continuous control. For discrete control, we use the MinAtar suite [Young and Tian, 2019] which consists of visually simpler versions of games from the Arcade Learning Environment (ALE) [Bellemare et al., 2013]. The goal of MinAtar is to provide a benchmark that does not require the vast amounts of compute needed for the full ALE evaluation protocol, which usually involves training for 200M frames across 10 runs per game. This can be prohibitively expensive for researchers and thereby the MinAtar benchmark reduces the barriers present in studying deep RL research. We include scores on 24 games from the ALE benchmark for a reduced number of steps to observe if there are signs of improvement when using ReLo over PER. We provide full training curves for each environment in the supplementary material. We compare ReLo with LaBER [Lahire et al., 2021] on the DeepMind Control Suite and MinAtar benchmarks using the codebase released by the authors.

In addition to the per environment scores, we report metrics aggregated across environments based on recommendations from Agarwal et al. [2021] in Fig. 1. We can see that across a diverse set of

tasks and domains, ReLo outperforms PER as a prioritization scheme, with higher IQM and lower optimality gap scores. This highlights the generality of ReLo.

### 4.2.1 DMC

In the continuous control tasks, Soft Actor Critic (SAC) [Haarnoja et al., 2018] is used as the base off-policy algorithm to which we add ReLo. SAC has an online and an exponential moving average target $Q$ network which we use to generate the ReLo priority term as given in Eq. 10. For comparison, we also include SAC with PER to showcase the differences in performance characteristics of PER and ReLo. The results are given in Figs. 1a. On 6 of the 9 environments, ReLo outperforms the baseline SAC as well as SAC with PER. This trend in performance is visible in the aggregated scores in Fig. 1a where ReLo has a higher IQM score along with a lower optimality gap when compared to SAC, SAC with PER and LaBER.

Additionally, to study the effect of stochastic dynamics in a non-gridworld setting, we conducted an experiment on stochastic versions of DMC environments. Specifically, we added noise sampled from $\mathcal{N}(0, \sigma^2)$ to the environment rewards during training. During evaluation episodes, no noise is added to the reward. This is similar to the stochastic environments used by Kumar et al. [2020]. For Quadruped Run, Quadruped Walk, Walker Run we used $\sigma = 0.1$, and for Walker Walk we used a higher level of noise ($\sigma = 1$) since there wasn't much change in the performance when using $\sigma = 0.1$ compared to deterministic version. The results are presented in Fig. 3 and Table 2. These experiments highlight how ReLo outperforms uniform sampling and PER in stochastic dynamics while also attaining lower TD error and reinforce our claim that ReLo can effectively handle stochasticity in environments.

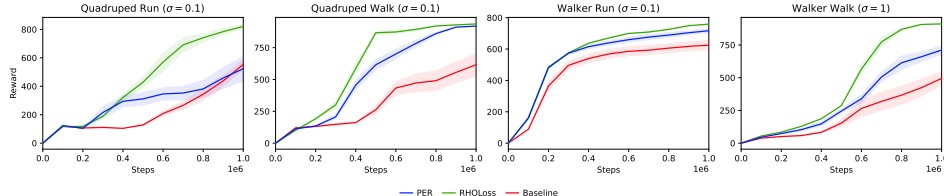

Figure 3: Performance on Stochastic DMC. Performance aggregated over 5 seeds.

Table 2: Validation TD Error on Stochastic DMC. Performance aggregated over 5 seeds.

|  | QUADRUPED RUN | QUADRUPED WALK | WALKER RUN | WALKER WALK |
|---|---|---|---|---|
| UNIFORM | 0.5 (0.34, 0.66) | 1.27 (1.07, 1.47) | 0.04 (0.04, 0.04) | 1.98 (1.96, 2.0) |
| PER | 5.22 (3.39, 7.05) | 0.53 (0.47, 0.59) | 0.06 (0.05, 0.07) | 3.62 (3.47, 3.78) |
| RELO | **0.19 (0.17, 0.2)** | **0.12 (0.11, 0.12)** | **0.03 (0.02, 0.03)** | **1.94 (1.92, 1.96)** |

### 4.2.2 OpenAI Gym Environments

In addition to the DeepMind Control Suite, we also evaluate ReLo on environments from the OpenAI Gym suite, namely HalfCheetah-v2, Walker2d-v2 and Hopper-v2 and the results are available in Fig. 1b. There is a general trend where PER leads to worse performance when compared to the baseline algorithm, in line with previous work by Wang and Ross [2019] that shows that the addition of PER to SAC hurts performance. However, this is not the case when using ReLo as a prioritization scheme. It is very clear how the PER degrades performance while ReLo does not adversely affect the learning ability of the agent and in fact leads to higher scores in each of the tested environments. We believe that the degraded performance of PER in the Gym environments is due to instability in training caused by rapidly varying value estimates. We provide experiments to back up this claim in Appendix L. We should that early terminations in the OpenAI Gym environments cause the PER agent to have the highest variance in the estimate of the fail state distribution. These results are presented in Table 3 When we disable early terminations, the PER agent is able to learn in these environments, though it is still worse than the ReLo agent.

Table 3: Variance in Value Estimate of Fail State Distribution.

| METHOD | MEAN (CIS) | MIN | MAX |
|---|---|---|---|
| BASELINE | 29.055 (-62.551, 120.662) | -87.8392 | 258.484 |
| PER | 149.982 (-76.382, 376.346) | -115.291 | 793.237 |
| RELO | 18.225 (-0.307, 36.756) | -14.4796 | 49.37 |

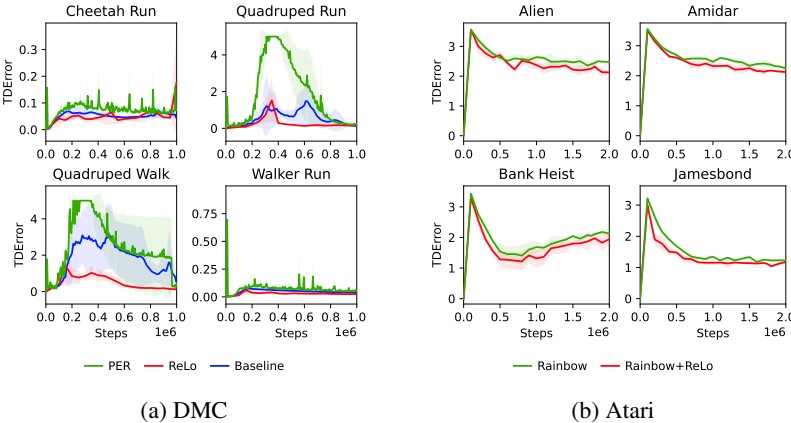

(a) DMC        (b) Atari

Figure 4: Comparison of training TD error for PER and ReLo based sampling in a) DMC and b) Atari benchmarks. Calculated over 5 seeds.

### 4.2.3 MinAtar

In the MinAtar benchmark, we use DQN [Mnih et al., 2015] as a baseline algorithm and compare its performance with PER and ReLo on the 5 environments in the benchmark. DQN does not have a moving average target $Q$ network and instead performs a hard copy of the online network parameters to the target network at a fixed interval. Similar to the implementation of ReLo in SAC, we use the online and target $Q$ network in the ReLo equation for calculating priorities. The results on the benchmark are given in Fig. 1c. PER performs poorly on Seaquest and SpaceInvaders, with scores lower than the baseline DQN. These results are consistent with observations by Obando-Ceron and Castro [2021] which analyzed the effect of the components of Rainbow in the MinAtar environment. In contrast, ReLo consistently outperforms PER and is comparable to or better than the baseline. Our previous observation that ReLo tends to help improve performance in situations where PER hurts performance is also true here.

### 4.2.4 ALE

As an additional test, we modified the Rainbow [Hessel et al., 2017] algorithm, which uses PER by default, to instead use ReLo as the prioritization scheme and compared it against Rainbow with PER on 24 environments from the ALE benchmark. Instead of the usual 200M frames of evaluation, we trained each agent for 2M frames to study if there are gains that can be observed in this compute-constrained setting. As shown in Fig. 1d, we see that Rainbow with ReLo achieves better performance than Rainbow with PER. These experiments show the versatility of ReLo as a prioritization scheme.

### 4.3 Analysis of TD Loss Minimization

To verify if using ReLo as a prioritization scheme leads to lower loss values during training, we logged the TD error of each agent over the course of training and these loss curves are presented in Fig. 4. As we can see, ReLo does indeed lead to lower TD errors, empirically validating our claims that using ReLo helps the algorithm focus on samples where the loss can be reduced. Another interesting point is that in Fig. 4a, SAC with PER has the highest reported TD errors throughout training. This is due to PER prioritizing data points with high TD error which might not be necessarily learnable. Data points with higher TD error are repeatedly sampled and thus making the overall

losses during training higher. In contrast, ReLo addresses this problem by sampling data points that are learnable and leads to the lowest TD errors during training.

Table 4: Comparison of validation TD Errors on DM Control Suite

|  | BASELINE | PER | RELO |
|---|---|---|---|
| CHEETAHRUN | $0.02 \pm 0.002$ | $0.03 \pm 0.003$ | $0.12 \pm 0.033$ |
| QUADRUPEDRUN | $0.62 \pm 0.055$ | $2.24 \pm 0.127$ | $0.35 \pm 0.067$ |
| QUADRUPEDWALK | $3.17 \pm 0.252$ | $2.11 \pm 0.189$ | $1.07 \pm 0.146$ |
| WALKERRUN | $0.08 \pm 0.015$ | $0.15 \pm 0.018$ | $0.06 \pm 0.008$ |

Table 5: Comparison of validation TD Errors on Atari

|  | RAINBOW | RAINBOW+RELO |
|---|---|---|
| ALIEN | $2.329 \pm 0.408$ | $2.331 \pm 0.251$ |
| AMIDAR | $2.157 \pm 0.120$ | $2.055 \pm 0.143$ |
| BANK HEIST | $2.044 \pm 0.222$ | $2.018 \pm 0.261$ |
| JAMESBOND | $1.653 \pm 0.819$ | $1.142 \pm 0.174$ |

**Validation TD Error:** We also compared the validation TD errors of each method after training in Tables 2, 4, and 5. This was done by collecting $10^4$ frames from the environment and computing the mean TD errors of these transitions. These results show that the validation TD errors [Li et al., 2023] obtained at the evaluation phase are actually lower for ReLo. This analysis is very similar to observations in [Li et al., 2023] where the authors find that lower validation TD error is a reliable indicator of good sample efficiency for off-policy RL algorithms. ReLo achieves lower validation TD error compared to uniform sampling and PER on DM Control Suite and Atari. When we study the correlation between validation TD error and performance in Appendix 11 and observe a strong correlation between the two metrics, confirming the findings of Li et al. [2023]. These results highlight another crucial reason for the robustness and good performance of ReLo across multiple domains.

## 5    Conclusion

In this paper, we have proposed a new prioritization scheme for experience replay, Reducible Loss (ReLo), which is based on the principle of frequently sampling data points that have the potential for loss reduction. We obtain a measure of the reducible loss through the difference in loss of the online model and a hold-out model on a data point. In practice, we use the target network in $Q$ value methods as a proxy for a hold-out model.

ReLo avoids the pitfalls that come with naively sampling points based only on the magnitude of the loss since having a high loss does not imply that the data point is actually learnable. While alleviating this issue, ReLo retains the positive aspects of PER, thereby improving the performance of deep RL algorithms. It is very simple to implement, requiring just the addition of a few lines of code to PER, and similar to PER it can be applied to any off-policy algorithm. Since it requires only one additional forward pass through the target network, the computational cost of ReLo is minimal, and there is very little overhead in integrating it into an algorithm.

While the reducible loss can be intuitively reasoned about and tested empirically, future work should theoretically analyze the sampling differences between ReLo and PER about the kind of samples that they tend to prioritize or ignore. This deeper insight would allow us to find flaws in how we approach non-uniform sampling in deep RL algorithms similar to work done in Fujimoto et al. [2020]. It would provide an analysis of the change in learning dynamics induced by PER and ReLo.

## Acknowledgments and Disclosure of Funding

The authors would like to thank the Digital Research Alliance of Canada for compute resources, NSERC, Google and CIFAR for research funding. We would like to thank Nathan Rahn and Pierluca D'Oro for their feedback and comments. We also thank Nanda Krishna, Jithendaraa Subramanian, Vincent Michalski, Soma Karthik and Kargil Mishra for reviewing early versions of this paper.

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

# A Implementation Details

We build our experiments on top of existing implementations of SAC, DQN and Rainbow. For the DeepMind Control Suite experiments, we modify Yarats and Kostrikov [2020], adding a prioritized replay buffer and the ReLo version. We use an open source implementation of Rainbow[2] for the Arcade Learning Environment and the DQN implementation from the MinAtar authors Young and Tian [2019]. Aside from the collected frames and number of seeds, we have not modified any of the hyper-parameters from these original implementations. The hyper-parameters as well as hardware and software used are given in Table 6.

Table 6: Hyper-Parameters of all experiments

| Environments | Algorithm | Algorithm Parameters | Hardware & Software |
|---|---|---|---|
| ALE | Rainbow | Frames = $2 \times 10^6$
seeds = 5

Remaining hyper-parameters same as Hessel et al. [2017] | Hardware-
CPU: 6 Intel Gold 6148 Skylake
GPU: 1 NVidia V100
RAM: 32 GB

Software-
Pytorch: 1.10.0
Python: 3.8 |
| DeepMind Control Suite | SAC | Frames = $1 \times 10^6$
seeds = 5

Remaining hyper-parameters same as Haarnoja et al. [2018] | Hardware-
CPU: 6 Intel Gold 6148 Skylake
GPU: 1 NVidia V100
RAM: 32 GB

Software-
Pytorch: 1.10.0
Python: 3.8 |
| MinAtar | DQN | Frames = $5 \times 10^6$
seeds = 5

Remaining hyper-parameters same as Mnih et al. [2015] | Hardware-
CPU: 6 Intel Gold 6148 Skylake
GPU: 1 NVidia V100
RAM: 32 GB

Software-
Pytorch: 1.10.0
Python: 3.8 |

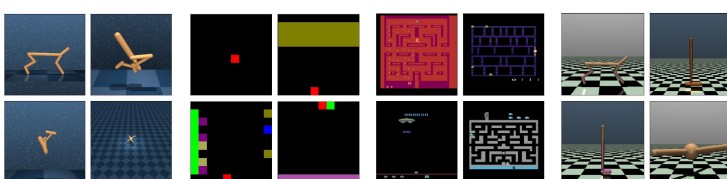

Figure 5: Visualization of a few environments from each benchmark. Left to right: DeepMind Control Suite, MinAtar, Arcade Learning Environment

# B Mapping Functions for ReLo

Prioritized experience replay buffers expect the priorities assigned to data points to be non-negative. While the MSE version of the TD error used in PER satisfies this constraint, ReLo does not. Therefore, there must be a non-negative, monotonically increasing mapping from ReLo to $p_i$. In our main experiments, we clipped negative ReLo values to zero. Another mapping we tried was to set $p_i = e^{ReLo}$, in which case the probability of sampling a data point $P_i$, from Eq. 6, corresponds to the softmax over ReLo scores. However, for this choice the priority would explode if the ReLo crossed

---

[2]https://github.com/Kaixhin/Rainbow

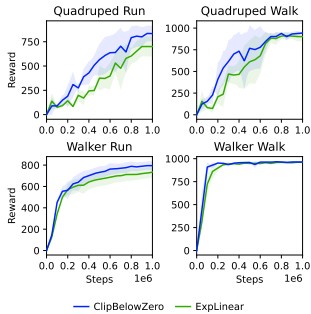

Figure 6: Comparison of different mapping functions from ReLo to $p_i$ on a subset of environments from the DMC benchmark. Performance is evaluated over 3 seeds.

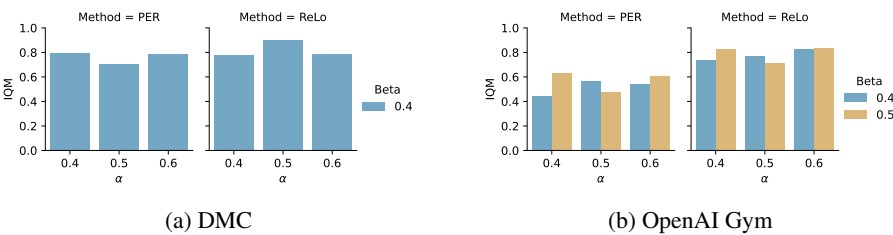

$$\text{(a) DMC} \qquad\qquad\qquad\qquad\qquad \text{(b) OpenAI Gym}$$

Figure 7: Sensitivity of ReLo and PER to $\alpha$ and $\beta$.

values above 40 which happened occasionally during the initial stages of learning in Rainbow. The second mapping function candidate was exponential when ReLo is negative and linear otherwise, that is,

$$f_{ExpLinear} = \begin{cases} e^{\text{ReLo}} & \text{if ReLo} < 0 \\ \text{ReLo} + 1 & \text{otherwise} \end{cases}$$

The linear part is shifted so that the mapping is smooth around ReLo = 0. As shown in Fig. 6, $f_{\text{ExpLinear}}$ performs worse compared to just clipping negative ReLo values to zero. When the ReLo values during training are analysed, we observe that the average of ReLo values (before the mapping) tends to be positive, so clipping does not lead to a large loss in information.

## C  Hyperparameter Sensitivity

We used the default hyperparameters that were given by Schaul et al. [2016], i.e $\alpha = 0.5$, $\beta = 0.4$ for all benchmarks for PER and ReLo. Rainbow [Hessel et al., 2017] used these values for the ALE benchmark after extensive grid search. Obando-Ceron and Castro [2021], which studied DQN and Rainbow on the MinAtar benchmark, used the same values for $\alpha$ and $\beta$ after hyperparameter tuning, hence we reused these values in our experiments. These values transferred well to the DMC domain and so we used them for all continuous control experiments too. We performed a hyperparameter sweep over $\alpha$ and $\beta$ for DMC and OpenAI gym and the results can be found in Fig 7. ReLo consistently outperforms PER in these sweeps highlighting the robustness of ReLo across benchmarks. These sweeps also show that the performance of ReLo varied with changing the value of $\alpha$ which implies that the priorities assigned by ReLo are not close to zero and hence the sampling behavior of ReLo is not uniform. If the priorities were close to zero, then performance would not change when we vary the value of $\alpha$ since Eq. 6 would then reduce to $p_{uniform}$ regardless of the value of $\alpha$.

## D  Extended Related Work

### D.1  Prioritization Schemes

Prioritization strategies have been leading to important improvements in sample efficiency. Sinha et al. [2020] proposes an approach that re-weights experiences based on their likelihood under the

stationary distribution of the current policy in order to ensure small approximation errors on the value function of recurring seen states. Lahire et al. [2021] introduces the Large Batch Experience Replay (LaBER) to overcome the issue of the outdated priorities of PER and its hyperparameter sensitivity by employing an importance sampling view for estimating gradients. LaBER first samples a large batch from the replay buffer then computes the gradient norms and finally down-samples it to a mini-batch of smaller size according to a priority.

Kumar et al. [2020] presents Distribution Correction (DisCor), a form of corrective feedback to make learning dynamics more steady. DisCor computes the optimal distribution and performs a weighted Bellman update to re-weight data distribution in the replay buffer. Inspired by DisCor, Regret Minimization Experience Replay (ReMERN) Liu et al. [2021] estimates the suboptimality of the Q value with an error network. Yet, Hong et al. [2022] uses Topological Experience Replay (TER) to organize the experience of agents into a graph that tracks the dependency between Q-value of states.

While PER was initially proposed as an addition to DQN-style agents, Hou et al. [2017] have shown that PER can be a useful strategy for improving performance in Deep Deterministic Policy Gradients (DDPG) Lillicrap et al. [2016]. Another recent strategy to improve sample efficiency was to introduce losses from the transition dynamics along with the TD error as the priority Oh et al. [2022]. Although this has shown improvements, it involves additional computational complexity since it also requires learning a reward predictor and transition predictor for the environment. Our proposal does not require training additional networks and hence is similar in computational complexity to PER. This makes it very simple to integrate into any existing algorithm. Wang and Ross [2019] propose an algorithm to dynamically reduce the replay buffer size during training of SAC so that the agent prioritizes recent experience while also ensuring that updates performed using newer data are not overwritten by updates from older data. However, they do not distinguish between points based on learn-ability and only assume that newer data is more useful for the agent to learn.

## D.2 Off-Policy Algorithms

Off-policy algorithms are those that can learn a policy by learning from data not generated from the current policy. This improves sample efficiency by reusing data collected by old versions of the policy. This is in contrast to on-policy algorithms such as PPO Schulman et al. [2017], which after collecting a batch of data and training on it, discard those samples and start data collection from scratch. Recent state-of-the-art off-policy algorithms for continuous control include Soft Actor Critic (SAC) Haarnoja et al. [2018] and Twin Delayed DDPG (TD3) Fujimoto et al. [2018]. SAC learns two $Q$ networks together and uses the minimum of the $Q$ values generated by these networks for the Bellman update equation to avoid over estimation bias. The $Q$ target update also includes a term to maximize the entropy of the policy to encourage exploration, a formulation that comes from Maximum Entropy RL Ziebart et al. [2008]. TD3 is a successor to DDPG Lillicrap et al. [2016] which addresses the overestimation bias present in DDPG in a similar fashion to SAC, by learning two $Q$ networks in parallel, which explains the "twin" in the name. It learns an actor network $\mu$ following Eq. 4 to compute the maximum over $Q$ values. TD3 proposes that the actor networks be updated at a less frequent interval than the $Q$ networks, which gives rise to the "delayed" name. In discrete control, Rainbow Hessel et al. [2017] combines several previous improvements over DQN, such as Double DQN van Hasselt et al. [2016], PER Schaul et al. [2016], Dueling DQN Wang et al. [2016], Distributional RL Bellemare et al. [2017] and Noisy Nets Fortunato et al. [2018].

# E  DeepMind Control Suite

We choose 9 environments from the DeepMind Control Suite Tassa et al. [2018] for testing the performance of ReLo on continuous control tasks. Each agent was trained on proprioceptive inputs from the environment for 1M frames with an action repeat of 1. The training curves for the baselines and ReLo are given in Fig. 8.

Table 7: Comparison of PER and ReLo on the DMC benchmark

|  | BASELINE | PER | LABER | RELO |
|---|---|---|---|---|
| CHEETAH RUN | $761.89 \pm 112.38$ | $\mathbf{831.87 \pm 38.90}$ | $579.80 \pm 60.61$ | $660.29 \pm 141.22$ |
| FINGER SPIN | $966.68 \pm 29.40$ | $975.40 \pm 6.75$ | $884.46 \pm 20.23$ | $\mathbf{978.78 \pm 14.46}$ |
| HOPPER HOP | $\mathbf{264.75 \pm 37.90}$ | $217.35 \pm 113.79$ | $90.85 \pm 57.76$ | $247.81 \pm 51.01$ |
| QUADRUPED RUN | $612.67 \pm 143.90$ | $496.42 \pm 216.01$ | $544.80 \pm 41.84$ | $\mathbf{833.92 \pm 81.05}$ |
| QUADRUPED WALK | $831.92 \pm 74.34$ | $766.30 \pm 200.86$ | $716.61 \pm 270.36$ | $\mathbf{942.64 \pm 9.75}$ |
| REACHER EASY | $\mathbf{983.06 \pm 2.70}$ | $981.58 \pm 6.33$ | $947.20 \pm 14.46$ | $979.08 \pm 11.02$ |
| REACHER HARD | $955.08 \pm 38.52$ | $935.08 \pm 47.94$ | $951.08 \pm 6.70$ | $\mathbf{956.80 \pm 38.73}$ |
| WALKER RUN | $759.13 \pm 23.91$ | $755.49 \pm 64.35$ | $551.81 \pm 58.41$ | $\mathbf{795.14 \pm 42.52}$ |
| WALKER WALK | $943.67 \pm 30.28$ | $957.38 \pm 8.24$ | $863.89 \pm 112.60$ | $\mathbf{963.28 \pm 5.03}$ |

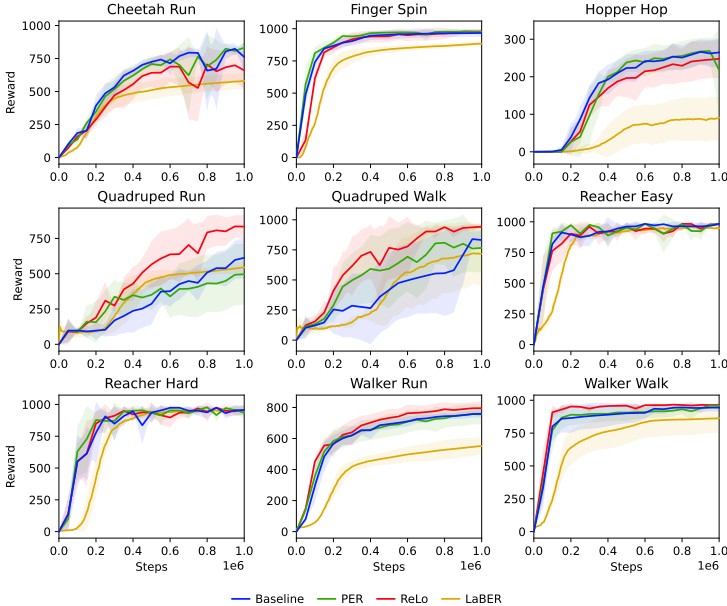

Figure 8: Training curves of environments from the DeepMind Control Suite. Performance is evaluated for 10 episodes over 5 random seeds.

## F    OpenAI Gym Environments

We evaluate agents for 1M timesteps on each environment and similar to DM Control, they are trained using proprioceptive inputs from the environment. The hyperparameters for this benchmark are shared with those used for the DM Control Suite experiments.

Table 8: Comparison of PER and ReLo on OpenAI Gym environments

|  | BASELINE | PER | RELO |
|---|---|---|---|
| GYM HALFCHEETAH | $9579.60 \pm 1331.00$ | $9549.42 \pm 917.92$ | $\mathbf{11590.63 \pm 670.36}$ |
| GYM HOPPER | $\mathbf{2795.18 \pm 659.19}$ | $2175.09 \pm 456.11$ | $2527.93 \pm 648.61$ |
| GYM WALKER | $2854.20 \pm 623.69$ | $735.16 \pm 474.89$ | $\mathbf{3514.39 \pm 676.80}$ |

## G    MinAtar

We evaluate the baselines against all 5 environments in the MinAtar suite Young and Tian [2019]. A visualization of a few environments from the suite is presented in Fig. 5. Each agent receives

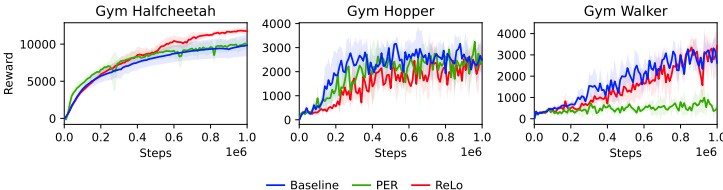

Figure 9: Training curves of environments from the OpenAI Gym benchmark. Performance is evaluated for 10 episodes over 5 random seeds.

the visual observations from the environment and is trained for 5M frames following the evaluation methodology outlined in Young and Tian [2019]. The training curves are given in Fig. 10.

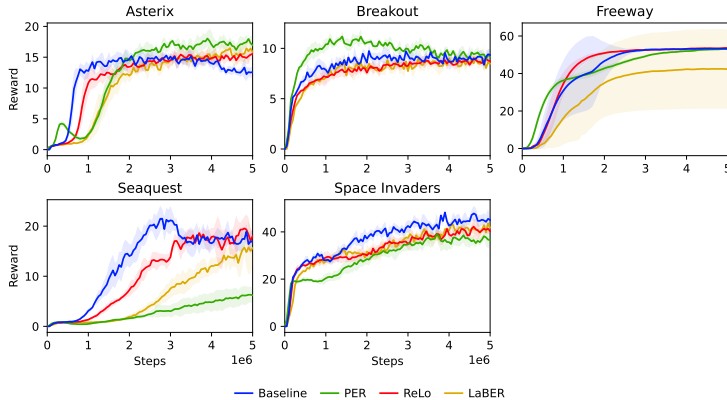

Figure 10: Training curves of environments from the MinAtar benchmark. Performance is evaluated using a running average over the last 1000 episodes over 5 random seeds.

Table 9: Comparison of PER and ReLo on the MinAtar benchmark

|  | BASELINE | PER | LABER | RELO |
|---|---|---|---|---|
| ASTERIX | $12.54 \pm 1.08$ | $\mathbf{16.16 \pm 1.02}$ | $15.51 \pm 1.11$ | $15.68 \pm 0.89$ |
| BREAKOUT | $\mathbf{9.36 \pm 0.29}$ | $8.88 \pm 0.72$ | $8.81 \pm 0.61$ | $8.98 \pm 0.75$ |
| FREEWAY | $52.80 \pm 0.35$ | $52.75 \pm 0.22$ | $41.98 \pm 20.99$ | $\mathbf{53.25 \pm 0.37}$ |
| SEAQUEST | $16.13 \pm 2.88$ | $6.02 \pm 1.92$ | $14.63 \pm 2.98$ | $\mathbf{18.13 \pm 1.25}$ |
| SPACE INVADERS | $\mathbf{45.36 \pm 1.65}$ | $37.36 \pm 4.45$ | $43.67 \pm 3.17$ | $38.54 \pm 2.60$ |

## H   Arcade Learning Environment

We evaluate agents on a compute-constrained version of the Arcade Learning Environment Bellemare et al. [2013], training each agent for 2M frames. We chose the standard 24 environments from the suite for our evaluation. ReLo is competitive with PER Schaul et al. [2016] in the tested environments. The training curves for the Temporal Difference Error and the rewards are given in Fig. 11 & Fig. 12 respectively.

## I   Gridworld

We implement a simple GridWorld for the experiments that highlights the drawbacks of PER with TD loss prioritization in Section 5.6. It consists of a $7 \times 7$ grid. A visualization of the grid is given in Fig. 13. The start state of the agent, represented by the blue point, is at the top left and the goal state is at the bottom right, represented by the green point. The agent is represented by the black point. The

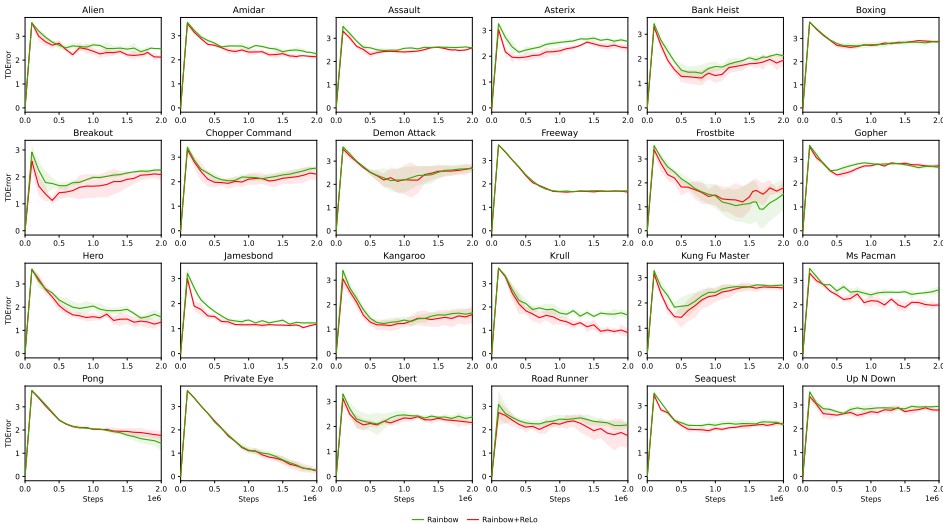

Figure 11: Temporal difference loss curves for Rainbow (with PER) and Rainbow with ReLo. Rainbow with ReLo achieves lower loss compared to PER, showing that ReLo is able to prioritize samples with reducible loss. The dark line represents the mean and the shaded region is the standard deviation over 3 seeds.

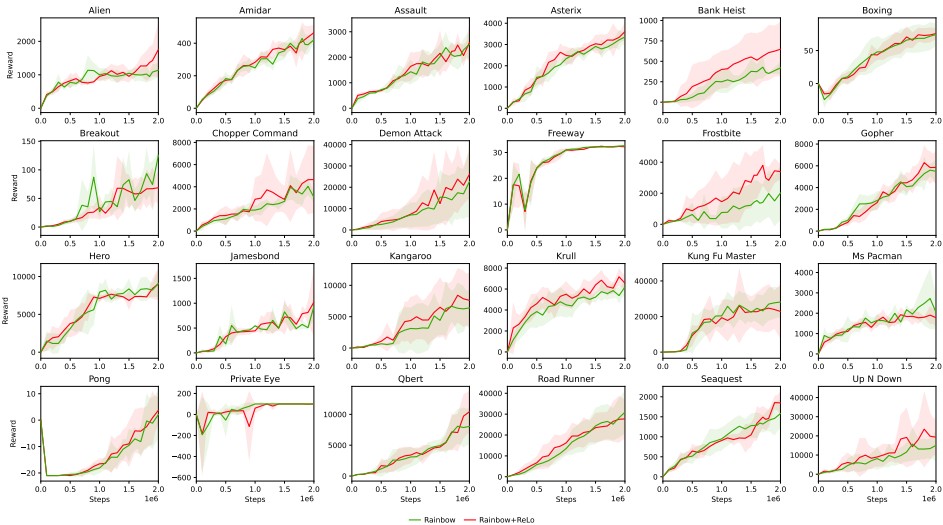

Figure 12: Training curves of 24 environments from the ALE benchmark. Performance is evaluated for 10 episodes over 5 random seeds.

agent gets a reward of $+2$ when it reaches the goal state, but does not receive reward anywhere else other than the stochastic point. The red point marks the state with a corresponding reward uniformly sampled from $[-0.5, 0.5]$. The locations of these points is fixed and does not change during training or evaluation. The state that the agent receives is $(x, y)$ where $x$ and $y$ are the coordinates of the agent's location, $x, y \in [-3, 3]$ and $x, y \in \mathbb{Z}$. The agent has 4 actions, to move up, down, left or right which will deterministically move the agent in that direction by 1 unit. If the agent is at the edge of the grid and takes an action that will move it out of the $7 \times 7$ grid, then it remains in the same location.

Table 10: Comparison of Rainbow with PER and Rainbow with ReLo on the ALE benchmark

|  | RAINBOW | RAINBOW W/ RELO |
| --- | --- | --- |
| ALIEN | 1278.70 ± 223.14 | **1352.90 ± 535.70** |
| AMIDAR | 376.20 ± 81.07 | **410.10 ± 85.63** |
| ASSAULT | 2241.78 ± 648.17 | **2617.37 ± 555.97** |
| ASTERIX | 3214.50 ± 323.70 | **3352.00 ± 431.98** |
| BANKHEIST | 526.80 ± 277.83 | **641.80 ± 284.67** |
| BOXING | **76.65 ± 14.58** | 76.21 ± 11.59 |
| BREAKOUT | **82.03 ± 46.10** | 68.36 ± 45.39 |
| CHOPPERCOMMAND | 2794.00 ± 732.87 | **4974.00 ± 2801.93** |
| DEMONATTACK | 25500.50 ± 16311.26 | **29294.30 ± 15905.25** |
| FREEWAY | **32.58 ± 0.31** | 32.35 ± 0.25 |
| FROSTBITE | 2850.60 ± 1553.34 | **3532.30 ± 1270.73** |
| GOPHER | 5336.60 ± 1023.75 | **5679.80 ± 1199.68** |
| HERO | **9907.15 ± 2108.71** | 8830.30 ± 1894.51 |
| JAMESBOND | 810.00 ± 412.00 | **902.00 ± 469.40** |
| KANGAROO | **8904.00 ± 3879.97** | 8091.00 ± 3618.47 |
| KRULL | 6553.18 ± 803.15 | **6718.67 ± 799.47** |
| KUNGFUMASTER | **29371.00 ± 8525.69** | 23654.00 ± 12360.40 |
| MSPACMAN | **2094.70 ± 614.58** | 1755.80 ± 374.32 |
| PONG | 3.18 ± 9.02 | **5.96 ± 6.45** |
| PRIVATEEYE | 100.00 ± 0.00 | 100.00 ± 0.00 |
| QBERT | 8382.00 ± 2935.16 | **10900.25 ± 2704.23** |
| ROADRUNNER | **29333.00 ± 9465.78** | 29222.00 ± 8696.91 |
| SEAQUEST | 1377.20 ± 362.57 | **1848.80 ± 788.85** |
| UPNDOWN | 17065.40 ± 7637.25 | **21241.50 ± 8599.97** |

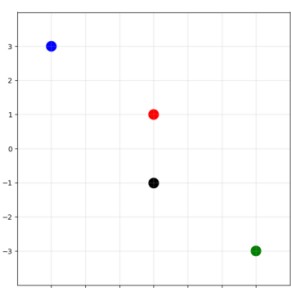

Figure 13: A top down view of the GridWorld. The agent is the black point. It starts at the blue point and the goal state is the green point. The red point represents the location of the stochastic reward.

## J  Forgetting Experiment

We visualize the gridworld used for the forgetting experiment in Fig. 14. The training curves are given in Fig. 15.

## K  Correlation between Validation TD Error and Policy Performance

We study the correlation between validation TD error and the performance of the policy and are able to reproduce the findings of Li et al. [2023], showing that there is indeed high correlation between the validation TD error and the performance. The results are in Table 11. We performed a paired t-test and bold entries where the results are significant.

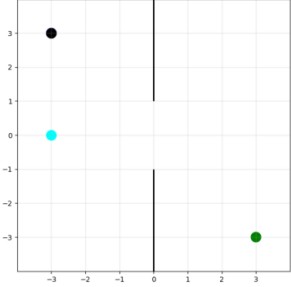

Figure 14: Visualization of Grid Room. Cyan is goal state in Room A, green is goal state in Room B. Goal A gives a reward of +1 and Goal B gives +5.

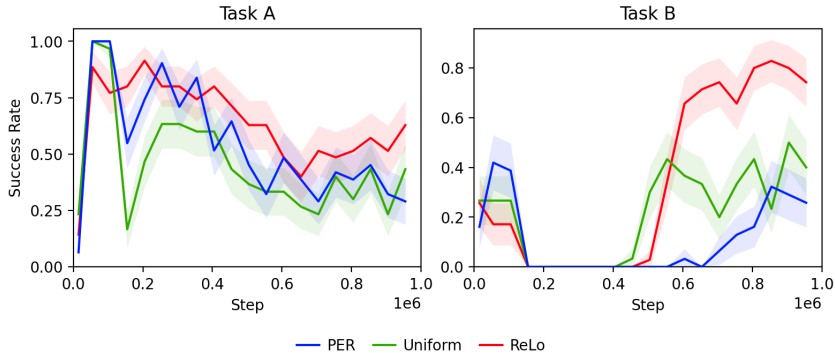

Figure 15: Performance curves of Uniform sampling, PER and ReLo on the forgetting task. Success rates are averaged over 60 seeds and shaded regions indicate standard deviation. Once the agent is no longer able to collect data about Task A (>100K steps), performance on the Task A decreases, but ReLo exhibits the least degradation in performance on Task A while still learning to solve Task B.

Table 11: Validation TD Error and Policy Performance on the DMC benchmark

| METHOD | TD ERROR$_{\text{BEST}}$ | RETURN$_{\text{BEST}}$ |
|---|---|---|
| CHEETAHRUN | PER | PER |
| FINGERSPIN | **RELO** | RELO |
| HOPPERHOP | **RELO** | BASELINE |
| QUADRUPEDRUN | **RELO** | **RELO** |
| QUADRUPEDWALK | LABER | **RELO** |
| REACHEREASY | **RELO** | BASELINE |
| REACHERHARD | BASELINE | RELO |
| WALKERRUN | **RELO** | **RELO** |
| WALKERWALK | **RELO** | **RELO** |

## L  Instability of PER in Mujoco Environments

We believe that degraded performance of the PER agent in Mujoco is because of instability in learning caused by rapidly varying value estimates. To test this hypothesis we studied the Walker2d environment, where PER obtains a mean reward of ~700, compared to the baseline (uniform sampling) which obtains a mean reward of ~2800. This is surprising since PER does learn to perform well in WalkerWalk, which is the DMC equivalent of Walker2d (they have similar agent morphologies).

An important difference in the two environments is that Walker2d has early termination (episode ends when the walker falls down) while WalkerWalk does not. This meant that the early terminations could make predicting the value of the fail state distribution difficult as the walker falls in different ways. There could be a lot of noise in the reward at that stage which can make the TD estimate noisy too.

We hypothesize that since PER only samples datapoints proportional to the TD error, these states would be repeatedly sampled as the noisy estimate would make the TD error high. This means it would be less likely to prioritize the samples corresponding to good behavior which could have lower TD error than the noisy fail states. We looked at the value estimate of the fail state in the baseline, PER and ReLo agent and observed this was the case. This is presented in Table 12 and estimates are calculated over 40 episodes.

Table 12: Variance in Value Estimate of Fail State Distribution.

| METHOD | MEAN (CIS) | MIN | MAX |
|---|---|---|---|
| BASELINE | 29.055 (-62.551, 120.662) | -87.8392 | 258.484 |
| PER | 149.982 (-76.382, 376.346) | -115.291 | 793.237 |
| RELO | 18.225 (-0.307, 36.756) | -14.4796 | 49.37 |

There is high variance in the predicted value of the fail state for PER, meaning that invariably the TD error for these points would be high. But further training on noisy points does not help and instead makes the problem worse, causing the value estimate to diverge. This can cause instabilities in training and potentially derail learning.

Finally we created a modified version of Walker2d without early terminations and PER acheives much better performance (Mean score of 1943.65 compared to 700.5 in the original Walker2d) in this environment, validating our hypothesis. Besides removing early termination, other parameters of the environment and the hyperparameters of the PER agent were the same in both experiments. We also looked at the value of the initial states (the environment is randomly initialized so there is an initial state distribution) in Table 13 and PER has higher variance in the predicted value even here.

Table 13: Variance in Value Estimate of Fail State Distribution.

| METHOD | MEAN | MIN | MAX |
|---|---|---|---|
| BASELINE | 227.916 (215.415, 240.417) | 215.929 | 252.337 |
| PER | 249.379 (177.265, 321.494) | 177.968 | 337.663 |
| RELO | 203.893 (195.357, 212.428) | 193.014 | 214.811 |

This analysis adds credence to our hypothesis that PER suffers from high variance in value estimates which hurt learning. Additionally, the experiments also show that ReLo has the least variance in the predicted value of the state (initial or fail state), highlighting how ReLo is a more stable prioritization scheme.

