# OpenReview forum: "Prioritizing Samples in Reinforcement Learning with Reducible Loss"
_NeurIPS.cc/2023/Conference — NeurIPS 2023 poster_

### Official Review · Reviewer_ax9T · 2023-07-02

**Soundness:** 4 excellent
**Presentation:** 3 good
**Contribution:** 3 good
**Rating:** 7
**Confidence:** 4

**Summary:**

This paper proposes an experience replay priority scheme based on the notion of reducible loss (ReLo), estimated as the difference in Q-loss between using the online Q-network and the target Q-network, and show in experiments in DM Control Suite, Mujoco tasks, and Atari environments that it improves upon prioritized experience replay (PER) (TD-error based) and uniform replay. They also demonstrate through a toy example that their method is able to avoid the common pitfall of stochasticity that prioritized experience replay struggles with often.

**Strengths:**

The strength of this paper is in its simplicity and clarity. The idea and implementation is very simple and straightforward, requiring no new changes to existing algorithms and networks, and takes advantage of the target network that is already present. The priority is very simple to compute, as it just requires computing the Q-loss one more time. The experiments are quite clear, and demonstrate how ReLo can outperform uniform sampling and PER across control/robotic tasks and MinAtar tasks. This all makes this idea very easy to adopt and further experiment with for follow-ups.

**Weaknesses:**

The main weakness is in the experiments. One weakness is in the ALE experiments, which do not run Rainbow for very long (only 2M). While these preliminary results are promising, they are ultimately too early in training to be relied upon. Nevertheless they are still useful to see than not to have. Another weakness is the lack of more stochastic domains, as control/mujoco and Atari are all very close to deterministic. Was sticky-actions enabled for Atari or MinAtar tasks? If not, then that would be something very good to include that would not require running on a new domain.

---- After Author Rebuttals ----

After reading other reviews and author rebuttals, I do agree that the experiments are a little lacking. There should be a clear (non-gridworld) stochastic domain that can highlight the difference between uniform, PER, and ReLo. The addition of noisy DMC results would fill this gap, and as the authors will include it in the final paper. I maintain my score.

**Questions:**

As mentioned in the weaknesses section, for the MinAtar and Atari experiments, did you enable sticky-actions? If yes, what was the sticky probability?

Because ReLo uses the target network to compute priorities, it is currently tied to the hyperparameters of the target network. Have you considered using a second target network to disentangle this? Could there be cases where for the RL, you would want a very fast updating target network, but for ReLo you would want a much slower updating target network?


**Limitations:**

This is related to the question about having a second target network. ReLo currently cannot easily change the hyperparameters for the target network, as it may have a great impact on the RL optimization. It may end up being the case that ReLo would prefer a much slower (or much faster) updating target network. This seems like a possible future direction to explore. Perhaps there should be a brief mention of this in the paper.

---

> ### Author Rebuttal · Authors · 2023-08-10
>
> We appreciate the reviewer's constructive criticism and recommendations for making this paper better. We discuss the points below.
>
> ### Sticky Actions
> Yes we used sticky actions in Atari and MinAtar, utilizing the default values of 0.25 and 0.1 respectively.
>
> ### Decoupling Target Network and Held Our Model
> We thank the reviewer for pointing this out. We agree it would be an interesting question to see if ReLo could be implemented with a separate target network. For our experiments we wanted to highlight the robustness and generality of ReLo by studying it with different target update mechanisms and target update frequencies and hence we did not use an additional target network.
>
> But we agree this would indeed be an interesting proposition. We performed some preliminary experiments with separate target networks and the performance was similar. However we are running some ablation experiments with faster and slower update frequencies.
>
> ### Additional experiments
> To highlight the ability of ReLo to mitigate forgetting, we created an experiment where an agent is given access to a sequence of tasks. This is explained in detail in the consolidated response. The experiment shows how the ReLo agent suffers the least degradation in performance on prior tasks while still learning on the new task. This is in contrast to the PER agent which forgets how to solve the previous task and also takes longer to learn the new task.

---

> > ### Comment · Reviewer_ax9T · 2023-08-14
> >
> > After reading the other reviews and author rebuttals, I think some of the points made in the other reviews make sense, such as MinAtar and DMC/Mujoco are perhaps not the best environments for a main experimental result. They are great as supporting results, but having a (non-gridworld) stochastic dynamics environment where PER clearly struggles. In light of this, I'm inclined to slightly lower my score, but I do still think this paper presents a clear, simple and interesting idea.

---

> > > ### Author Response · Authors · 2023-08-17
> > >
> > > Thank you for pointing this out. While we appreciate the concerns raised by you and the other reviewers, we would like to point out that MinAtar and Atari already have stochastic dynamics due to the presence of sticky actions. We also highlight that PER suffers significantly in MinAtar and is worse than the uniform sampling baseline in this benchmark. ReLo on the other hand does better than the baseline and PER as evident by the higher IQM scores. In Atari too, there is an increase in performance when utilizing ReLo as the prioritization scheme when compared to PER.
> > >
> > > However, to provide further evidence, we also conduct an additional study on stochastic versions of DMC environments. Specifically, we added noise sampled from $\mathcal{N}(0, \sigma^2)$ to the environment rewards during training. During evaluation episodes, no noise is added to the reward. This is similar to the stochastic environments used by Kumar et al. 2020 [1].
> > >
> > > We chose a random subset of the DMC suite given the time constraint, choosing Quadruped Run, Quadruped Walk, Walker Run and Walker Walk. We are running the entire suite and can add the results to the revised version of the paper. The results of this experiment after 500K steps and 1M steps are presented below. For Quadruped Run, Quadruped Walk, Walker Run we used $\sigma = 0.1$, and for Walker Walk we used a higher level of noise ($\sigma=1$) since there wasn't much change in the performance when using $\sigma=0.1$ compared to deterministic version. The tables clearly show the sample efficiency of ReLo, with it having higher performance than the baselines even early in training. Results are calculated over 5 seeds.
> > >
> > > ### 500K
> > > |         | Quadruped Run $\sigma = 0.1$          | Quadruped Walk $\sigma = 0.1$        | Walker Run $\sigma = 0.1$             | Walker Walk $\sigma = 1$            |
> > > |:--------|:------------------------|:------------------------|:------------------------|:------------------------|
> > > | PER     | 311.35 (262.91, 359.79) | 615.69 (569.33, 662.06) | 639.24 (621.78, 656.71) | 245.41 (223.57, 267.24) |
> > > | ReLo    | **428.76 (389.24, 468.28)** | **867.15 (850.01, 884.29)** | **670.22 (663.87, 676.56)** | **287.18 (284.04, 290.33)** |
> > > | Uniform | 128.04 (118.93, 137.15) | 262.71 (228.01, 297.42) | 568.65 (536.16, 601.15) | 153.48 (120.94, 186.02) |
> > >
> > > ### 1M Steps
> > > |         | Quadruped Run $\sigma = 0.1$          | Quadruped Walk $\sigma = 0.1$         | Walker Run $\sigma = 0.1$             | Walker Walk $\sigma = 1$            |
> > > |:--------|:------------------------|:------------------------|:------------------------|:------------------------|
> > > | PER     | 523.66 (433.24, 614.07) | 919.14 (914.74, 923.54) | 716.75 (700.55, 732.94) | 711.5 (676.09, 746.92)  |
> > > | ReLo    | **821.16 (800.2, 842.13)**  | **936.26 (932.04, 940.47)** | **759.28 (754.95, 763.62)** | **911.19 (907.23, 915.14)** |
> > > | Uniform | 553.91 (514.67, 593.16) | 616.06 (524.01, 708.1)  | 625.14 (592.51, 657.78) | 495.18 (441.55, 548.82) |
> > >
> > > We presented the stochastic gridworld setting in Section 4.6 to
> > > - Showcase a drawback caused by PER in stochastic environments
> > > - Show how ReLo is able to handle stochasticity by prioritizing points relevant to the main task over 'unlearnable' points (the point with stochastic reward).
> > >
> > > We would also like to draw your attention to the gridworld setting that was suggested by Reviewer DXpG to study task switches. We show how ReLo can mitigate the effect of forgetting while PER and uniform sampling show higher levels of degradation in performance. Based on feedback from Reviewer DxPG, we ran the experiment for longer, training for 1M environment steps and more seeds. There is now minimal overlap between the confidence intervals of the different methods.
> > >
> > > | Algorithm   | Task A            | Task B            |
> > > |:------------|:-----------------:|:-----------------:|
> > > | PER         | 0.29 (0.19, 0.39) | 0.26 (0.16, 0.36) |
> > > | Uniform     | 0.43 (0.32, 0.54) | 0.40 (0.29, 0.51) |
> > > | ReLo        | 0.63 (0.52, 0.74) | 0.74 (0.65, 0.84) |
> > >
> > > In our paper, we motivate ReLo by highlighting the pitfalls of TD error prioritization through gridworld environments. Our main experimental results on the DMC, MinAtar, Mujoco and Atari benchmarks are to highlight the versatility of ReLo prioritization in 1) varied domains (pixel and proprioceptive), 2) control schemes (continuous and discrete) and 3) TD update mechanisms (EMA and hard copy updates). We would also like to emphasize that our goal is to have an alternative stable prioritization scheme which addresses issues in PER.
> > >
> > > Thank you again for your insightful feedback which improves the quality of our work. We believe these additional experiments address your concerns about the validity of our results and would kindly request you to increase the score. We are also happy to address any other concerns.
> > >
> > > [1] DisCor: Corrective Feedback in Reinforcement Learning via Distribution Correction

---

> > > > ### Comment · Reviewer_ax9T · 2023-08-17
> > > >
> > > > Thanks for pointing to the stochastic dynamics of sticky-actions. However I do want to mention what reviewer DXpG brought up which was that sticky-actions adds the same noise to all actions at every timestep, which is not the most interesting form of noise. The smaller gridworld experiments do a better job of demonstrating more complex stochastic dynamics. Furthermore, as I mentioned in my review, your Atari experiments are a little to early to make strong conclusions, since Rainbow was not designed with Atari 100K in mind.
> > > >
> > > > The additional experiments in noisy DMC are a good addition, and do bolster the results. In light of this, I am willing to raise my score back.

---

### Official Review · Reviewer_Mxrp · 2023-07-06

**Soundness:** 3 good
**Presentation:** 2 fair
**Contribution:** 2 fair
**Rating:** 7
**Confidence:** 4

**Summary:**

The authors propose a novel form of prioritized experience replay based on the “learn-ability” of the sample, or how much the training loss on a sample can be reduced.
They point out that previous methods that prioritize data with high TD error are not optimal if the high TD error is irreducible error, which can be caused by a noisy environment.  Instead the authors prioritize high reducible TD error using a new metric that is the difference in TD error between the current Q function and a delayed, target Q function.  This is a simple and computationally cheap method that can be applied to many off-policy RL algorithms.
They demonstrate across a range of continuous and discrete control benchmarks that this sampling metric outperforms uniform and PER sampling and demonstrate that their method achieves lower TD error.

**Strengths:**

- Authors identify a valid limitation of prior methods: that it might over-emphasize samples with high reducible error leading to wasted training effort.
- Reasonable application of supervised learning techniques to RL to address this limitation in a relatively simple and straightforward algorithm.  Method also inherits generality of other prioritized experience replay algorithms.
- Authors demonstrate strong experimental results across a wide range of tasks and using multiple base RL algorithms.

**Weaknesses:**

- Clarity in writing.  Background and related work section can be more concise.  In section 3.1, the introduction of $f_{map}$ with PER priority is not very intuitive.  Some terms like "significance" and "importance" of data are not well defined, and "learn-ability" and reducible loss seem to be the same.

- In general, the experimental results are missing learning curves, especially since sample efficiency would be one of the main benefits of prioritized replay.  It’s also not clear how the final performance figures are determined (Are all methods trained until convergence?  How are the policies evaluated?).

- Section 4.5 TD Error Analysis.  The interpretation of the TD errors seems a bit stretched.  Many factors could affect TD error that does not necessarily relate to policy performance.  For example, that PER actually prioritizes high TD error samples like the authors mention.  Were the validation data gathered by the specific policy trained in each algorithm?  Were these policies trained until convergence?

**Questions:**

- Section 3 makes a nice connection between the hold-out model in supervised and the target Q network in RL.  Why isn’t the same reasoning applied to the hold-out data?  If the ReLo is computed on data that has been used to train the target network already, isn’t this no longer the reducible loss?

- Section 3.1: How are the priority values updated across the replay buffer as training progresses?  If the priority values are only updated per training mini-batch, is it possible that there are a large proportion of out-of-date priority values, especially if ReLo was low to begin with but increases later like in the forgetting case that was mentioned.

- Why is PER in most cases, worse than the baseline method?

- Section 4.6:  This is a nice toy analysis and may also be useful to mention earlier on in the paper to support the hypothesis that PER is limited due to unlearnable samples.  The issue is the learning curves for this task aren’t very convincing because of the high variance and that average performance never really goes above 0.5 for any methods.

- Why did you use LaBER as another comparison and why were the other prioritized replay methods not relevant to compare against?

**Limitations:**

No limitations addressed

---

> ### Author Rebuttal · Authors · 2023-08-10
>
> We thank the reviewer for their valuable insights and suggestions to improve the clarity of the paper. First, we would like to apologize for the lack of explanation and coherence in the paper. We have fixed all the points raised in the review. We will address them sequentially.
>
> ### Notation Clarification
> $f_{map}$ is used to denote the mapping from the raw ReLo values to a priority that can be used for sampling in the Priority Queue implementation of the PER buffer. We shall add clarifications in the revised version of the paper to highlight this.
>
> ### Significance, Importance and Learnability
> These are qualitative terms that we use to describe the phenonmenon that all data samples are not equally important for the learning process. We use learnability as a  the qualitative measure of how useful a sample is for the learning process. The reducible loss is a quantitative metric that can be used to measure the learnability of a sample.
>
> ### How final performance is determined
> We follow the recommendations of Agarwal et al. (2021) for aggregating results across a benchmark. They propose treating the performance of a given training run as a random variable and suggest that authors report statistical measures on these random variables. The interquartile mean (IQM) computes the mean of the middle 50% of runs while the optimality gap is a measure of how far an algorithm is from optimal performance aggregated across environments. For computing these measures, each environment score first needs to be normalized. In the DMC benchmark, the optimal score for each environment is 1000, while we use the highest reported scores for each environment from the MinAtar paper and the OpenAI Gym environments for calculating the optimality gap for those benchmarks. The environment scores are normalized with respect to this max score. For the ALE benchmark, we normalize the scores of each game with respect to reported random and human level scores.
>
> ### Learning Curves
>
> The learning curves for the experiments are given in the supplementary material.
>
>
>
> ### Relationship between validation TD Error and Policy Performance
> We have addressed this in the Common Author Response. To summarize, there is indeed a correlation between the validation TD error and the policy performance in the trained off policy agents.
>
> ### Calculation of Validation TD Error
> Yes, the validation data was collected by the same policy after training was completed. The policies are trained for 1M steps (for DMC and Mujoco, which is standard for these benchmarks) or 2M steps (Atari).
>
> ### Separate Hold-out Data
> One approach would be to collect a subset of trajectories from the environment during the training process to create an evolving held out dataset. These trajectories would be only in the held out buffer and not the training buffer. The main RL agent would learn from the training buffer, and parallely we could learn a new Q network on only the held-out buffer to mimic the held out model. However this process would consume additional computational resources. In general, since the samples collected are non i.i.d, target networks can be a good approximation for this held out model.
>
> ### Updating Priority Values
> The priorities are updated for the batch of samples that are trained on. So the priorities in the buffer are the last encountered priorities of the sample. This is based on the implementation used by PER since it can be very expensive to recalculate the priorities of all samples in the buffer after every update. The addition of $\epsilon$ to the probabilities ensures that all samples are replayed and if there is an erroneous priority it would be corrected the next time the data point is sampled.
>
> ### Why PER does worse than uniform in certain environments?
> In a function approximation setting, it is generally quite difficult to obtain the correct values of every state accurately. When an agent enters a region of the state space that it has not thoroughly explored before, it is bound to get a spike in the TD Error for all these samples and PER will repeatedly sample these new states. This might make the slightly older states non-consistent because we are in the function approximation setting and thus this can lead to forgetting. ReLo prevents this by sampling states based on the reducible loss and thus prevents forgetting.
>
>
> ### Learning curves in Section 4.6
> Section 4.6 motivates why PER is not a good prioritization scheme and how ReLo would be able to handle these cases. We did not train those agents for longer because we are more interested in the sampling strategies of the baselines and how they handle stochasticity in environments.
>
> ### Why LaBER was used as a baseline?
> LaBER did not require training any additional networks to obtain better prioritization schemes and required similar compututational resources to Prioritized Experience Replay. That is why, we think LaBER and PER would be fair baselines since ReLo also does not involve training any additional networks which can be expensive.

---

> > ### Comment · Reviewer_Mxrp · 2023-08-18
> >
> > Thank you for your response.  This has addressed my main concerns about clarity and about how experimental results are reported and I have updated my score accordingly.

---

### Official Review · Reviewer_DXpG · 2023-07-06

**Soundness:** 3 good
**Presentation:** 3 good
**Contribution:** 2 fair
**Rating:** 6
**Confidence:** 3

**Summary:**

This work introduces a novel experience replay sampling technique (ReLo-based sampling) that prioritizes sampling experience that has the greatest potential for reducing the agent’s loss. Empirically, ReLo-based sampling yield better performance than uniform random sampling and prioritized experience replay (PER).

**Strengths:**

1. The work tackles a known issue with PER,
2. The writing is generally very clear, and the authors consider a diverse set of benchmark tasks (discrete/continuous control, visual vs state-based observations).

**Weaknesses:**

1. The authors discuss how ReLo-based sampling can help in tasks with stochastic dynamics and/or rewards and help prevent forgetting, though it seems like the chosen benchmark tasks don't illustrate all of these benefits.
    * DMC and OpenAI Gym environments have deterministic dynamics and rewards.
    * MinAtar has stochastic dynamics (sticky actions with probability 0.1), but this stochasticity is completely independent of the agent’s state and chosen action. Thus, this stochasticity makes every state “equally unlearnable” in a sense.
    * In contrast, I do see how ReLo is helpful if a subset of states have stochastic dynamics/rewards as in the toy example in section 4.6.

Does this mean the observed benefits of ReLo can then be attributed to how it prevents forgetting?

2. I would like to see further discussion on how ReLo helps prevent forgetting, especially if the reasoning in my first comment is valid. Currently, there is one sentence mentioning this benefit. If the authors can provide a toy example similar to the gridworld example in section 4.6, that would more clearly illustrate why ReLo is important. Something that comes to mind: Consider an agent that must simultaneously learn 2 different tasks (e.g. navigate to point A and point B in a gridworld task). The first half of training, the agent sees task 1. The second half, it sees task 2. Presumably, ReLo would prevent the agent from forgetting task 2 while learning task 1?

3. I believe section 4.6 should be moved just before ReLo-based sampling is introduced; it concretely illustrates a problem with PER and demonstrates how ReLo-based sampling addresses it.

4. Related to my two previous comments: I feel ReLo needs more motivation. Section 4.6 does a good job of motivating one aspect of ReLo, though I would really like to see motivation for the other aspects (forgetting, stochastic rewards).

I am willing to raise my score if the authors can (1) clarify which benefits of ReLo the experiments highlight, (2) better motivate ReLo e.g. through the use of additional tasks that highlight each potential benefit of ReLo.

**Questions:**

1. Can the authors include the returns for the tasks shown in Figure 2? Does large TD-error correspond to poor performance for these tasks? Since LaBER addresses issues with PER, It would be informative to see the TD-error for LaBER in Figure 2 as well.

**Limitations:**

See Weaknesses.

---

> ### Author Rebuttal · Authors · 2023-08-10
>
> We thank the reviewer for their valuable comments and suggestions to improve the paper. We strongly believe that these suggestions would make the paper better. We address all the concerns and questions below.
>
> ### How ReLo can help even without stochasticity in rewards or dynamics
> Yes, this can be attributed to how it prevents forgetting. We provide ReLo's motivation by illustrating how prioritising the TD error might be problematic when the reward or dynamics are stochastic (Section 4.6). However, ReLo can also help in preventing forgetting. In the function approximation setting, it is generally quite difficult to obtain the correct values of every state accurately because a change in the network affects the values of the entire state space. When an agent enters a region of the state space that it has not explored before, it is bound to get a spike in the TD error for all these new samples and PER will repeatedly sample these states. This might make the slightly older states non-consistent because we are in a function approximation setting. ReLo would prevent this by allowing the agent to still sample based on the reducible loss, so it would only prioritize learnable samples.
>
> ### How ReLo can prevent forgetting
> We thank you for this suggestion and we performed a similar experiment where we indeed confirmed that ReLo can actually prevent forgetting. We elaborate on this and share intereseting insights in the Common Author Response.
>
> ### Section 4.6
> We shall move this section in the revised version of the paper.
>
> ### Benefits of ReLo
> In this paper we study the effect of the ReLo criterion on
> - handling noisy/stochastic points (Section 4.6)
> - mitigating forgetting of previous tasks (Forgetting experiment in the common response)
>
> We also highlight its
> - generality and robustness to mechanism and frequency of target network updates (EMA/Hard-copy target update)
> - applicability to varied tasks with discrete or continuous control, with pixel or proprioceptive inputs.
>
> ### Returns for experiments in Figure 2
> The returns are given in Table 2 in the supplementary material.
>
> ### Relationship between validation TD Error and Policy Performance
> We have addressed this in the Common Author Response. To summarize, there is indeed a correlation between the validation TD error and the policy performance in the trained off policy agents.
>
> ### TD Errors for LaBER
> We included the validation TD Error for LaBER in the Common Author Response. We will add the traning TD Error in the revised version of the paper.

---

> > ### Comment · Reviewer_DXpG · 2023-08-15
> >
> > Thank you for the additional experiments and clarification on the benefits of ReLo. I do still have a few concerns, and currently maintain my score.
> >
> > In the common response pdf, the success rates in Fig. 2 don't seem to be statistically significant; he standard deviations overlap quite a bit. Can the authors perform a paired t-test at a 95% confidence level for ReLo vs. PER and ReLo vs. Uniform to determine if these results are significant? It may also help to increase the training budget. I have these same concern for Fig 3. in the main paper; While ReLo avoids sampling a state with low learnability, it's unclear if ReLo's return is any better than than PER or Uniform.
> >
> > Overall, the results show that ReLo produces smaller TD errors during training (Fig. 2, Fig. 6 in appendix), though it's unclear if this reduction improves performance. In Fig. 3 of the appendix, ReLo clearly improves data efficiency in 3/9 tasks (quadruped run, quadruped walk, walker walk) and maybe in walker walk. In Fig. 4 of the appendix, it improves data efficiency only in HalfCheetah, and in Fig. 5, there's no obvious improvement offered by ReLo. I suspect many of the return results in Table 1 of the common response pdf are not statistically significant; could you also provide significance tests for this table and, for instance, highlight/bold cells where ReLo outperforms the others with significance?

---

> > > ### Author Response · Authors · 2023-08-18
> > >
> > > We thank the reviewer for this additional feedback and concerns regarding statistical significance. We address their concerns sequentially below.
> > >
> > > ### Forgetting Experiment
> > > We increased the training budget to 1M environment steps (keeping the budget for task A constant at 100K) and 60 seeds, and report the results below. There is minimal overlap between the confidence intervals now and ReLo still shows the least degradation in performance compared to the baselines.
> > >
> > > | Algorithm   | Task A            | Task B            |
> > > |:------------|:-----------------:|:-----------------:|
> > > | PER         | 0.29 (0.19, 0.39) | 0.26 (0.16, 0.36) |
> > > | Uniform     | 0.43 (0.32, 0.54) | 0.40 (0.29, 0.51) |
> > > | ReLo        | 0.63 (0.52, 0.74) | 0.74 (0.65, 0.84) |
> > >
> > >
> > > ### Pitfalls of TD Error Prioritization (Figure 3)
> > > While the variance can also be reduced with a higher training budget and more seeds, the main attempt of Figure 3 was not to show the improved performance of ReLo in terms of rewards but to highlight cases where TD Error prioritization fails due to stochasticity in the environment due to poor sampling (the bar chart in Figure 3 which is statistically significant). We agree that running it with higher training budget will increase readabilty of the figure. These experiments are currently running and we will include it in the revised version of the paper.
> > >
> > > ### Significance of Table 1 in Common Response
> > >
> > > We thank the reviewer for pointing this out. We performed a paired t-test of the the returns and the validation TD error in DMC. For the return, the following environments where ReLo does better are statistically significant, Quadruped Run, Quadruped Walk, Walker Run and Walker Walk. For the validation TD errors, all the environments where ReLo has better validation TD are significant. Accordingly we have made bold the corresponding rows in the TD error - Return correlation table and present it below. We will also update the paper accordingly.
> > >
> > > | Method         | $TD_{Best}$   | $Return_{Best}$   |
> > > |:---------------|:----------|:--------------|
> > > | CheetahRun    | PER       | PER           |
> > > | FingerSpin    | **ReLo**      | ReLo          |
> > > | HopperHop     | **ReLo**      | Baseline      |
> > > | QuadrupedRun  | **ReLo**      | **ReLo**          |
> > > | QuadrupedWalk | LaBER     | **ReLo**          |
> > > | ReacherEasy   | **ReLo**      | Baseline      |
> > > | ReacherHard   | Baseline  | ReLo          |
> > > | WalkerRun     | **ReLo**      | **ReLo**          |
> > > | WalkerWalk    | **ReLo**      | **ReLo**          |

---

> > > > ### Author Response · Authors · 2023-08-18
> > > >
> > > > ### Stochastic experiments
> > > >
> > > > > The authors discuss how ReLo-based sampling can help in tasks with stochastic dynamics and/or rewards and help prevent forgetting, though it seems like the chosen benchmark tasks don't illustrate all of these benefits.
> > > >
> > > > We thank the reviewer for raising this point. To study the effect of stochastic dynamics in a non-gridworld setting, we conducted an experiment on stochastic versions of DMC environments. Specifically, we added noise sampled from $\mathcal{N}(0, \sigma^2)$ to the environment rewards during training. During evaluation episodes, no noise is added to the reward. This is similar to the stochastic environments used by Kumar et al. 2020 [1].
> > > >
> > > > We chose a subset of the DMC suite given the time constraint, choosing Quadruped Run, Quadruped Walk, Walker Run and Walker Walk. We are running the entire suite and can add the results to the revised version of the paper. The results of this experiment after 500K steps and 1M steps are presented below. For Quadruped Run, Quadruped Walk, Walker Run we used $\sigma = 0.1$, and for Walker Walk we used a higher level of noise ($\sigma=1$) since there wasn't much change in the performance when using $\sigma=0.1$ compared to deterministic version. The tables clearly show the sample efficiency of ReLo, with it having higher performance than the baselines even early in training. Results are calculated over 5 seeds.
> > > >
> > > > #### 500K
> > > > |         | Quadruped Run $\sigma = 0.1$          | Quadruped Walk $\sigma = 0.1$        | Walker Run $\sigma = 0.1$             | Walker Walk $\sigma = 1$            |
> > > > |:--------|:------------------------|:------------------------|:------------------------|:------------------------|
> > > > | PER     | 311.35 (262.91, 359.79) | 615.69 (569.33, 662.06) | 639.24 (621.78, 656.71) | 245.41 (223.57, 267.24) |
> > > > | ReLo    | **428.76 (389.24, 468.28)** | **867.15 (850.01, 884.29)** | **670.22 (663.87, 676.56)** | **287.18 (284.04, 290.33)** |
> > > > | Uniform | 128.04 (118.93, 137.15) | 262.71 (228.01, 297.42) | 568.65 (536.16, 601.15) | 153.48 (120.94, 186.02) |
> > > >
> > > > #### 1M Steps
> > > > |         | Quadruped Run $\sigma = 0.1$          | Quadruped Walk $\sigma = 0.1$         | Walker Run $\sigma = 0.1$             | Walker Walk $\sigma = 1$            |
> > > > |:--------|:------------------------|:------------------------|:------------------------|:------------------------|
> > > > | PER     | 523.66 (433.24, 614.07) | 919.14 (914.74, 923.54) | 716.75 (700.55, 732.94) | 711.5 (676.09, 746.92)  |
> > > > | ReLo    | **821.16 (800.2, 842.13)**  | **936.26 (932.04, 940.47)** | **759.28 (754.95, 763.62)** | **911.19 (907.23, 915.14)** |
> > > > | Uniform | 553.91 (514.67, 593.16) | 616.06 (524.01, 708.1)  | 625.14 (592.51, 657.78) | 495.18 (441.55, 548.82) |
> > > >
> > > >
> > > > Additionally we looked at the TD errors of these experiments since the improved performance is statistically significant. As we can see, in all four environments, ReLo acheives the best TD error too. The TD errors are calculated at the end of training using the final checkpoint of the model.
> > > >
> > > > |         | Quadruped Run           | Quadruped Walk    | Walker Run        | Walker Walk       |
> > > > |:--------|:------------------------|:------------------|:------------------|:------------------|
> > > > | PER     | 5.22 (3.39, 7.05) | 0.53 (0.47, 0.59) | 0.06 (0.05, 0.07) | 3.62 (3.47, 3.78) |
> > > > | ReLo    | **0.19 (0.17, 0.2)**        | **0.12 (0.11, 0.12)** | **0.03 (0.02, 0.03)** | **1.94 (1.92, 1.96)** |
> > > > | Uniform | 0.5 (0.34, 0.66)        | 1.27 (1.07, 1.47) | 0.04 (0.04, 0.04) | 1.98 (1.96, 2.0)  |
> > > >
> > > > These experiments highlight how ReLo outperforms uniform sampling and PER in stochastic dynamics while also attaining lower TD error. They complement the experimental results from the main paper and reinforce our claim that ReLo can effectively handle stochasticity in environments.
> > > >
> > > > Thank you again for your insightful feedback which improves the quality of our work. We believe these additional experiments address your concerns about the validity of our results and would kindly request you to increase the score. We are also happy to address any other concerns.
> > > >
> > > > [1] [DisCor: Corrective Feedback in Reinforcement Learning via Distribution Correction]

---

> > > > > ### Comment · Reviewer_DXpG · 2023-08-19
> > > > >
> > > > > Thank you for your response! I'm satisfied with the forgetting experiment; it is now clear that ReLo protects against forgetting here. Also, the stochastic DMC results are interesting, showing that ReLo lowers TD error and improves performance. I have raised my score from Borderline Accept to Weak Accept.
> > > > >
> > > > > I do want to point out a few things that should be addressed in revisions:
> > > > >
> > > > > 1. In addition to moving Section 4.6 to an earlier part of the paper, I think the additional forgetting experiment you've provided should go here as well. These toy problems help immensely with understanding the two core purposes of ReLo (prevent forgetting, help with stochastic reward/dynamics).
> > > > >
> > > > > 2. Previously, we discussed that the primary benefit of ReLo in the DMC tasks without stochastic reward/dynamics is how it prevents forgetting on a smaller timescale (e.g. when the agent enters a state it hasn't seen before, it will experience a spike in TD error, and PER will repeatedly sample this experience). I think this "small scale forgetting" should be explicitly discussed as well. In particular, it would be very helpful to mention that the "forgetting" that happens in your toy experiment happens at a much smaller scale with PER in the DMC tasks (and presumably happens with PER in any RL task when using function approximation).
> > > > >
> > > > > 3. Since there's currently only one task explicitly showing how ReLo can help prevent forgetting experiment, the authors should include more. There are plenty of possibles, e.g.
> > > > >
> > > > >     * Use the DMC/MuJoCo suite and then let task A = move forward, and then task B = move backward.
> > > > >     * Use the 2D locomotion tasks in the DMC/MuJoCo suite and then let task A = move to point A, and then task B = move to point B.
> > > > >     * Use the Fetch [1] or Panda [2] robot arm tasks, and let task A = move block to goal A, and then task B = move block to goal B. Alternatively, you could do Task A = goal state is sampled from 1/2 of the goal space, and then task B = goal state is sampled from the other half of the goal space.
> > > > >
> > > > > 4. If accepted, I think readers will find the paper much more convincing if you can squeeze your additional rebuttal results into the main body of the paper rather than an appendix.
> > > > >
> > > > > [1] Plappert et. al. Multi-Goal Reinforcement Learning: Challenging Robotics Environments and Request for Research. arXiv:1802.09464
> > > > >
> > > > > [2] Gallouedec et. al. Panda-gym: Open-source goal-conditioned environments for robotic learning. arXiv:2106.13687

---

### Official Review · Reviewer_hZPB · 2023-07-07

**Soundness:** 3 good
**Presentation:** 2 fair
**Contribution:** 3 good
**Rating:** 5
**Confidence:** 3

**Summary:**

The author proposes a simple modification on prioritized experience replay, preventing unlearnable samples from being prioritized. Prioritized experience replay (PER) prioritizes samples with high TD errors, but some samples that have irreducible TD errors due to stochasticity may also be prioritized. This will hurt learning since these samples are unlearnable. Thus, the author proposes to prioritize only learnable samples by using reducible losses as priorities. The reducible losses are measured by the difference of TD errors with respect to the online Q-network and target Q-network. The experimental results show that the proposed method demonstrates consistent improvements over PER and uniform sampling in various common benchmarks.

**Strengths:**

- The idea of prioritizing reducible is a critical insight on prioritizing experience replay.
- The method is easy to implement
- The performance improvement seems to be consistent, while I have a few concerns on experimentation details.

**Weaknesses:**

- Lack of justification on why the difference between online and target networks can be a good estimate of reducible loss. A low online network loss may not indicate the Q-function learns the right Q-values since online networks may be changing too fast and cannot provide stable value targets.
- Experiments in Atari need more justification. Since not every Atari game is tested, it is required to show the reason of choosing these games. Do these games cover all types of games? Are they particularly challenging for PER? Also, the reason of training only 2M frames makes the experiments weak. It's understandable that training 200M frames is not attainable for most people. But I think it's good to think of why training 2M frames would be sufficient to tell the difference between PER (Rainbow) and ReLO. It seems that the performance of ReLO and PER in Atari is pretty close. For now, the results in Atari is a bit inconclusive. It's possible that in these compute-restricted settings, ReLO and PER won't have significant differences.

Overall, I think the insight made in this paper is important. However, the current experimental results do not clearly demonstrate why this insight matters since the performance is not significantly improved. In Atari, it's quite close to Rainbow (which is PER). In other domains, ReLO is doing better than PER but quite close to baselines (uniform sampling). To strengthen this paper, I suggest the author looking for tasks that significantly show the performance gain of ReLO to other tasks.

**Questions:**

- Why PER degrades in mujoco (openai gym) but not in deepmind control suite?
- In Table 1, ReLO doesn't have lower validation TD errors than PER and baselines in CheetahRun. Can the author comment on why?

**Limitations:**

Not mentioned in the paper. I suggest the author think of limitations when the target network is not a good held-out set and how to construct a better held-out set. Also, it's good to provide some theoretical analysis to show why the chosen priority will improve the convergence rate to the correct Q-function or similar analysis made in LaBER.

---

> ### Author Rebuttal · Authors · 2023-08-10
>
> We thank the reviewer for the useful suggestions and comments to help improve this paper. We have incorporated the recommendations into the paper and discuss them below.
>
> ### Motivation of ReLo in RL and target networks
> In Mindermann et al. (2022) the hold out model is trained on a validation dataset which is drawn from the same distribution as the training dataset. Infact, the implementation uses a subset of the training set to train the validation model. However in RL we deal with non stationarity data distributions so it is not possible to train a hold-out model. However, the problem of learnability of a sample still exists in RL. If we observe that the TD error has been high for a long time, then it might be the case that at those points there is inherent noise (e.g. receives random reward every time), making such points highly unlearnable. PER would emphasize such points, whereas ReLo will not. Thus, a lagging version of the online model (target network) can be used to capture such points.
>
> We believe target networks can be a good approximation for hold out model for two reasons:
> - Even though both models are trained on the same transitions, as explained earlier, there can be significant change in the data distribution that the online model sees that the target model does not till the next time the parameters are copied. As the policies approach close to optimal behavior, the difference in policy behavior between the online and the target network might reduce. However, as Schaul et al. (2022) mentions, the policies keep changing even when close to convergence, which means at no point are the distributions for training the online and target model same.
> - This is just an approximation for the holdout model, and we just use it as a prioritization scheme. So even the points that have low priority are still sampled. Thus in cases that the approximation may not hold good, it won’t affect the training in a major way as those points will still be sampled.
>
>
> ### Atari Experiments
> We have addressed your concerns in the Consolidated Author Response.
>
> ### Additional examples of the benefits of ReLo
> We have added another example to show how ReLo can prevent forgetting when switching to a new task. In addition, in our experiments in the paper, we show that ReLo is more robust and can be widely applied to many domains whereas PER only has limited benefits in particular environments.
>
> ### Poor Performance of PER in DMC
>
> From our inspection of the training process, we believe this could be due to instabilities in training caused by high TD errors. During training of the PER agent the TD error would sometimes explode and not recover.
>
> ### Higher Validation TD Error on CheetahRun
>
> ReLo does not outperform in the baselines in this particular task. It could be that prioritizing the TD error works really well in this environment which is why PER is the best performing algorithm.
>
> ### Limitations
> We will add a separate limitations paragraph in the Conclusions section in the revised version of the paper.
> * Separate held out Set instead of using Target Networks: One approach would be to collect a subset of trajectories from the environment during the training process to create an evolving held out dataset. These trajectories would be only in the held out buffer and not the training buffer. The main RL agent would learn from the training buffer, and in parallel, we could learn a new Q network on only the held-out buffer. However this process would consume additional computational resources. In general, since the samples collected are non i.i.d, target networks can be a good approximation for this held out model.
>
> *    Theoretical Analysis: We agree it would be interesting to perform a theoretical analysis of the change in training dynamics and convergence rate introduced by sampling with ReLo. We have mentioned this as an avenue for future work in the conclusion of the paper.

---

> > ### Comment · Reviewer_hZPB · 2023-08-18
> >
> > Thanks for the author's response. The responses answered my questions and justified two of my main concerns: why use target network and why test in Atari100k environments. However, the response didn't answer the rest of the questions (see questions). Also, the new experiments in gridworld are not well motivated. Why ReLo prevents forgetting is not explained. Does forgetting happen in other environments? A clear explanation and analysis of why the proposed method is also important for a good paper. That being said, since I believe that avoiding prioritizing data with irreducible loss is worth being presented to the community, I increased my rating to 5.

---

> > > ### Author Response · Authors · 2023-08-19
> > >
> > > We would like to thank the reviewer for the thoughtful discussions and reconsidering their score. We would like to address the further concerns raised in your comment.
> > >
> > > ### Why PER is bad in Mujoco
> > >
> > > Thank you for pointing this out. We now have a strong answer validated by our experiments. We believe that degraded performance in Mujoco is because of instability in learning caused by rapidly varying value estimates. To test this hypothesis we studied the Walker2d environment, where PER obtains a mean reward of ~700, compared to the baseline (uniform sampling) which obtains a mean reward of ~2800. This is surprising since PER does learn to perform well in WalkerWalk, which is the DMC equivalent of Walker2d (they have similar agent morphologies).
> > >
> > > An important difference in the two environments is that Walker2d has early termination (episode ends when the walker falls down) while WalkerWalk does not. This meant that the early terminations could make predicting the value of the fail state distribution difficult as the walker falls in different ways. There could be a lot of noise in the reward at that stage which can make the TD estimate noisy too.
> > >
> > > We hypothesize that since PER only samples datapoints proportional to the TD error, these states would be repeatedly sampled as the noisy estimate would make the TD error high. This means it would be less likely to prioritize the samples corresponding to good behavior which could have lower TD error than the noisy fail states. We looked at the value estimate of the fail state in the baseline, PER and ReLo agent and observed this was the case. These are calculated over 40 episodes.
> > >
> > > | Method   | Mean (CIs)                |       Min |     Max |
> > > |:---------|:-------------------------:|:---------:|:-------:|
> > > | Baseline | 29.055 (-62.551, 120.662)  |  -87.8392 | 258.484 |
> > > | PER      | 149.982 (-76.382, 376.346) | -115.291  | 793.237 |
> > > | ReLo     | 18.225 (-0.307, 36.756)    |  -14.4796 |  49.37  |
> > >
> > > There is high variance in the predicted value of the fail state for PER, meaning that invariably the TD error for these points would be high. But further training on noisy points does not help and instead makes the problem worse, causing the value estimate to diverge. This can cause instabilities in training and potentially derail learning.
> > >
> > > Finally we created a modified version of Walker2d without early terminations and PER acheives much better performance (Mean score of 1943.65 compared to 700.5 in the original Walker2d) in this environment, validating our hypothesis. Besides removing early termination, other parameters of the environment and the hyperparameters of the PER agent were the same in both experiments.
> > >
> > > We also looked at the value of the initial states (the environment is randomly initialized so there is an initial state distribution) and PER has higher variance in the predicted value even here.
> > >
> > >
> > > | Method   | Mean                 |     Min |     Max |
> > > |:---------|:--------------------------:|:-------:|:-------:|
> > > | Baseline | 227.916 (215.415, 240.417) | 215.929 | 252.337 |
> > > | PER      | 249.379 (177.265, 321.494) | 177.968 | 337.663 |
> > > | ReLo     | 203.893 (195.357, 212.428) | 193.014 | 214.811 |
> > >
> > > This analysis adds credence to our hypothesis that PER suffers from high variance in value estimates which hurt learning. Additionally, the experiments also show that ReLo has the least variance in the predicted value of the state (initial or fail state), highlighting how ReLo is a more stable prioritization scheme.
> > >
> > > ### Why ReLo has worse validation TD error in Cheetah Run
> > >
> > > ReLo also achieves lower returns than the baselines in Cheetah Run, and worse return usually corelates with worse validation TD error. Although the degradation in performance is not as much and all algorithms manage to solve the task.

---

> > > > ### Author Response · Authors · 2023-08-19
> > > >
> > > > ### Motivating how ReLo prevents forgetting
> > > >
> > > > We motivate ReLo by illustrating how prioritizing the TD error might be problematic when the reward or dynamics are stochastic (Section 4.6). Additionally, ReLo can help in preventing forgetting. In the function approximation setting, it is generally quite difficult to obtain correct values for every state accurately because a change in the network affects the values for the entire state space. When an agent enters a region of the state space that it has not explored before, the TD error is very likely to spike up for samples collected and thus, PER will repeatedly sample these states. This might result in making the value estimates poorer for older states that do not get sampled, as we are in the function approximation setting. ReLo would prevent this by allowing the agent to still sample based on the reducible loss, so it would only prioritize learnable samples.
> > > >
> > > > ### Motivating the new forgetting experiment
> > > >
> > > > We wanted to show how ReLo can mitigate forgetting by replaying samples which had low TD error previously but have higher TD error now (essentially the increased TD error implies the model forgot about these points). This would lead to a high ReLo as online TD would be high but target TD would be lower. This effect can mitigate forgetting and we designed a concrete and interpretable example to test this hypothesis. After 100k samples, the agent no longer can collect data about Task A, and can only continue to do well on it if it replays Task A samples from the buffer. So this setup can effectively test forgetting. And from our experimental results, ReLo produces the least degradation in performance in Task A while also learning quickly on Task B (we have the highest performance on Task A and B)
> > > >
> > > > ### Does forgetting happen in other experiments
> > > >
> > > > The other benchmarks don't have this explicit Task A & Task B structure, but yes you can "forget" the accurate value of a state even if you did well on it previously. This can happen in any function approximation setting, as explained earlier. ReLo would sample the older points more frequently than PER since the ReLo for forgotten points would be high. Hence this behavior would reduce the effect of forgetting.
> > > >
> > > > We believe that we have addressed your concerns now and would be grateful if you would consider increasing your score after reviewing our response. We would be happy to address any other concerns if time permits.

---

### Author Rebuttal · Authors · 2023-08-10

Thank you for taking the time to review our work. We greatly appreciate your valuable feedback and comments and we will incorporate the suggestions into the revised version of the paper. We would like to clarify a few points that multiple reviewers mentioned before addressing each comment individually.

### ReLo can help prevent forgetting
We created a 6x6 gridworld consisting of two rooms and a goal state in each room. The left room is called Room A and the right is Room B. We define Task A as the agent starting in the Room A and reaching the goal state in Room A, and similarly define Task B as the agent starting in Room B and reaching the goal in Room B. There is a single gap in the wall allowing the agent to explore both rooms, but a timelimit is implemented such that the agent can not reach both goals in one episode. For the first 100k environment steps, the agent begins in Room A and we block access to Room B. So during this stage, the agent only learns about Task A.

After 100K steps, the agent starts in Room B, allowing it to learn Task B. The agent no longer starts at Room A, thereby no longer collecting data about Task A and must retain its ability by replaying the relevant transitions from the buffer. We train three agents, a baseline DQN agent, a PER DQN agent and a ReLo DQN agent and monitor their performance on both tasks during training. We evaluate performance over 50 seeds, providing the average success rates at the end of training in the table below. The training curves and a visualization of the environment are included the PDF included in the general response.

| Task A       | Task B       |
| -----------  | -----------  |
|PER: 0.40     |PER: 0.64     |
|Uniform: 0.32 |Uniform: 0.72 |
|ReLo: 0.80    |ReLo: 1.00    |


From the training curves, we can see that there is a clear drop off in performance on Task A after 100K steps when the agent can no longer actively collect data on the task. However the ReLo agent exhibits least degradation in Task A while also outperforming the baseline and PER agent on Task B. This experiment clearly shows how ReLo helps the agent replaying relevant data points that could have been forgotten.

### Relationship between Validation TD Error and Performance (DXpG, Mxrp)
We analyzed the validation TD error and the return for environments in the DMC suite and observed that there is indeed a correlation between the two metrics. Of the 9 games tested, in 5 games the method with the best (lowest) validation TD error is also the method with the best (highest) policy performance. There is a similar correlation between the worst validation TD error and the worst performing policy.

### Atari Experiments (ax9T, hZPB, DXpG)
The subset of games were chosen from the games used in the Atari100K benchmark. This is a benchmark used for evaluating sample efficiency in Atari, where agents are provided with a budget of only 100K interactions with the environment. The suite was chosen to cover a range of games where non random performance can be obtained in 100K data regime. Hence we believe this subset would be a good suite of games to validate our hypothesis that ReLo improves sample complexity.

---

### Author Response · Authors · 2023-08-21
**Thanking everyone for the insightful discussions**

We are very grateful for the engaging and fruitful discussions. They have added significant value to the manuscript. We are working on incorporating additional experiments and comments that came up during the rebuttal and discussion to the revised version. We thank you again for your valuable time and feedback.

---

### Decision · Program_Chairs · 2023-09-21

**Decision:**

Accept (poster)

**Comment:**

This paper identified an issue of prioritized experience replay (PER), i.e., large TD error could be a result of irreducible noise, which makes PER problematic in stochastic transition and reward settings. The authors proposed to used a reducible loss based on target networks to do prioritization.

The idea of prioritizing reducible is appreciated by all reviewers and is a valid issue of PER. Reviewers also like the simple and direct proposed method, using target network to approximate reducible losses.

There are concerns regarding the experiments were conducted on nearly deterministic environments of DMC and OpenAI Gym, and the authors used a noisy DMC environment to provide additional evidences, which convinced reviewers.

Discussions and suggestions from reviewers, including forgetting experiments and improving writing should be taken into subsequent versions.